**Measurement report: The influence of particle number size distribution and hygroscopicity on the microphysical properties of cloud droplets at a mountain site**

Xiaojing Shen[1], Quan Liu[1,*], Junying Sun[1,*], Wanlin Kong[2], Qianli Ma[3], Bing Qi[2], Lujie Han[3], Yangmei Zhang[1], Linlin Liang[1], Lei Liu[1], Shuo Liu[1], Xinyao Hu[1], Jiayuan Lu[1], Aoyuan Yu[1], Huizheng Che[1], Xiaoye Zhang[1]

[1] State Key Laboratory of Severe Weather & Key Laboratory of Atmospheric Chemistry of CMA, Chinese Academy of Meteorological Sciences, Beijing, 100081, China

[2] Hangzhou Meteorological Bureau, Hangzhou, 310051, China

[3] Lin'an Atmosphere Background National Observation and Research Station, Lin'an 311307, Hangzhou, China

Corresponding to: Quan Liu (liuq@cma.gov.cn) and Junying Sun (jysun@cma.gov.cn)

**Abstract**: An automatic switched inlet system, incorporating a ground-based counterflow virtual impactor (GCVI) and a $PM_{2.5}$ impactor, was developed and employed to investigate the particle number size distributions (PNSDs) and chemical composition for cloud-free (CF), cloud interstitial (CI) and cloud residual (CR) particles at Mt. Daming in the Yangtze River Delta, China, throughout a one-month period in spring 2023. The PNSDs of CF particles were primarily characterized by a significant Aitken mode alongside a secondary accumulation mode. In contrast, CI and CR particles exhibited unimodal distribution with Aitken and accumulation modes, peaking at 56 and 220 nm, respectively. With the fast changes of PNSDs during the onset stage of the observed four typical cloud processes, it can be inferred that the critical diameters activated as cloud droplets ranging from 133 to 325 nm. Particularly noteworthy was the higher hygroscopicity parameter, $\kappa$ value observed in CR particles ($0.32\pm0.06$), associated with a larger mass fraction of nitrate, compared to the lower $\kappa$ value in CI particles ($0.23\pm0.08$), with higher fraction of black carbon. For a typical cloud process, the hygroscopicity of CI particles was found to influence cloud droplet properties, with higher $\kappa$ values corresponding to lower droplet number concentration, reduced liquid water content and smaller effective cloud droplet diameters. This suggests that these CI particles are capable of absorbing ambient water vapor, thereby restricting further droplet growth. This investigation contributes to understanding aerosol-cloud interactions by assessing the impact of aerosol particles on cloud microphysics, thus enhancing overall comprehension of these complex atmospheric dynamics. However, it's noted that long-term observations are necessary to capture more cloud processes and yield statistically significant findings.

# 1. Introduction

Aerosol-cloud interactions (ACI) still contribute to uncertainties in radiative forcing substantially (IPCC, 2021). Atmospheric aerosols serve as cloud condensation nuclei (CCN) and ice nucleating particles (INPs), impacting cloud formation and thereby solar radiation (Haywood and Boucher, 2000). They can suppress precipitation as the amount of solar radiation reaching the land surface was decreased, and enhance precipitation by accelerating the conversion of cloud water by cloud seeding (Rosenfeld et al., 2008). Aerosols may also enter cloud droplets or ice crystals through impaction with these particles (Pruppacher and Klett, 1997). The physical and chemical characteristics of aerosols and clouds are fundamental for understanding aerosol-cloud interactions (Köhler, 1936; Seinfeld and Pandis, 2016). Especially the particle size and chemical composition (determining the hygroscopicity) both matter the activation ability of the aerosols (Dusek et al., 2006). Aerosol particles undergo complex physical and chemical transformations during cloud formation (Pöschl, 2005). In the cloud passage investigated through airborne measurements, it has been found the nitrate entered the cloud droplets and governed by the gas-phase mass transfer process, whereas much of the sulfate in the cloud water is the result of nucleation scavenging (Hayden et al., 2008). In the in-situ measurement of fog events in Po Valley, Italy, the nucleation scavenging efficiency of inorganics species was 60-70%, and 40-50% for organics and black carbon (Gilardoni et al., 2014). Although the fog scavenging processes include impaction and nucleation scavenging, generally, nucleation scavenging process is dominant (Seinfeld and Pandis, 2016). Feedbacks among cloud macrophysics, radiation and turbulence can further influence the cloud microphysics, particularly in polluted conditions (Shao et al., 2023). Sulfate and nitrate formation through aqueous-phase chemistry during cloud processes can alter the mixing state and activation potential of resuspended aerosols (Yao et al., 2021). Cloud droplets can capture soluble particles (e. g. sulfate, salts), non-soluble particles (e. g. black carbon, biological aerosol particles), gases, and other cloud droplets through nucleation and impaction collisions and coalescence (Pereira Freitas et al., 2024; Zieger et al., 2023). Upon precipitation formation, these materials are removed from the atmosphere via wet deposition. However, if the precipitation does not occur and cloud droplets evaporates instead, dissolved materials and some insoluble components in residual particles can modify the particle mixing states, sizes and even the chemical compositions (Hoose et al., 2008).

There are few studies on the physical and chemical properties of cloud interstitial and residual particles have been conducted by applying ground-based counterflow virtual impactor (GCVI) in recent years. Utilizing GCVI coupled with a chemical ionization mass spectrometer, research conducted at the Zeppelin

Observatory in Ny-Ålesund, Svalbard, revealed that sulfuric acid made a more significant contribution to cloud residuals than aerosols during cloud-free conditions (Gramlich et al., 2023). In remote Arctic regions, Aitken mode particles were found to play a significant role in cloud droplet formation, especially when accumulation mode particle concentrations were low (Karlsson et al., 2022; 2021). Additionally, the effects of cloud processes on particle mixing-state, hygroscopicity, black carbon, and coarse-mode/bioaerosols were also discussed by using the same GCVI in Zepplin (Adachi et al. 2022, Duplessis et al. 2024, Zieger et al. 2023, Pereira Freitas et al. 2024). Satellite observations have revealed a non-linear relationship between cloud droplet number concentration ($N_d$) and fine mode aerosol concentration (as indicated by Aerosol Index), as the $N_d$ sensitivity decreases with the high aerosol loading, leading to a challenge in accurately assessing cloud droplet changes influenced by aerosols (Jia and Quaas, 2023). To improve the understanding of aerosol-cloud interactions through both observational and modelling approaches, in-situ measurements of aerosol and cloud variables at mountain sites are particularly helpful and essential.

In China, numerous studies have been conducted to investigate the physico-chemical and optical properties of cloud droplets, encompassing both cloud residual and interstitial particles at mountain sites, utilizing a GCVI in conjunction with other aerosol measurement instruments (Guo et al., 2022; Zhang et al., 2017a; Zhang et al., 2017b). However, discussions concerning the dynamic processes of cloud formation, maturity, and dissipation, as well as the influence of aerosols on cloud formation, have been limited. To enhance understanding of the microphysical characteristics of aerosol-cloud-precipitations, we conducted aerosol and cloud droplet measurements using an automatic switched inlet system coupled with a GCVI and PM$_{2.5}$ cyclone, alongside a series of instruments, at a high-altitude station in economical developed and densely-populated Yangtze River Delta (YRD) region in China. This study aims to elucidate the comprehensive characteristics of cloud-free, cloud interstitial and cloud residual particles, and to uncover how variations in particle number size distribution and hygroscopicity (chemical composition) impact their potential for activation as cloud droplets in the ambient environment of supersaturated conditions.

## 2. Measurements and Methodology

### 2.1 Observation site

The measurements were conducted from April 12 to May 8 in 2023 at a meteorological Radar Station situated at Mt. Daming (30.03°N, 118.99°E, 1470 m, above sea level, asl) in Hangzhou, Zhejiang Province,

which located in Yangtze River Delta (YRD) region of China. Mt. Daming is characterized by a terrain of lower hills and mountains, with peak heights below 1500 m asl., covering an area of approximately 24.7 km$^2$. The surrounding landscape includes densely polluted lowland areas extending northeastward. Designated as a national geopark, Mt. Daming is enveloped by forests, attracting tourists primarily from the south side of the mountain for sightseeing, with only few people hiking to the nearby of the observation site. The observatory is located approximately 120 km southwest of the urban areas of Hangzhou, belonging to the YRD region, and it typically resides within the planetary boundary layer (PBL) during daytime and within the free troposphere (FT) during nighttime, facilitating the investigation of air mass exchanges between the PBL and FT. Furthermore, it offers an opportunity to assess the transport of well-mixed air masses originating from the YRD region, encompassing megacities such as Hangzhou, Shanghai, Nanjing, etc. The figure of observation site and surrounding environment is given as Fig. S1 in Supplementary Materials (SM).

## 2.2 Instrumentation

An automatic three-way switched inlet system was developed, comprising a PM$_{2.5}$ cyclone and a GCVI (model 1205, Brechtel Manufacturing Inc., USA). Aerosols were sampled through a PM$_{10}$ impactor and a PM$_{2.5}$ cyclone (Thermo Fisher Scientific, USA) in sequence, with a flow rate of 16.7 L min$^{-1}$. But the actual total flow rate passing through the inlet system was 14.5 L min$^{-1}$, including a Twin Scanning Mobility Particle Sizer (TSMPS, TROPOS, Germany) with flow rate of 3.5 L min$^{-1}$, a Mixing Condensation Particle Counter (MCPC, model 1720, Brechtel Manufacturing Inc., USA), 1.0 L min$^{-1}$, an Aerodyne High Resolution Time-of-Flight Aerosol Mass Spectrometer (HRToF-AMS, Aerodyne Research, Inc., USA), 0.5 L min$^{-1}$, a cloud condensation nuclei counter (CCNc-100, DMT Inc., Boulder, CO, USA), 0.5 L min$^{-1}$, and a Nephelometer (Model 9003, TSI, USA) connected in series with MAAP (Multi-Angle Absorption Photometer, Model 5012, Thermo Fisher Scientific, USA), with the flow rate of 9.0 L min$^{-1}$. This inlet system was controlled by two magnetic ball valves embedded in the PM$_{2.5}$ cyclone line and the GCVI line, respectively. During cloud-free conditions, ambient air was sampled via the PM$_{2.5}$ inlet and passed through an automatic regenerating absorption aerosol dryer to maintain the relative humidity (RH) below 30% (Tuch et al., 2009), with the valve state marked as 0. In this study, the RH of aerosol from PM$_{2.5}$ inlet system was 24.3±6.4% as measured by a RH sensor included with the TSMPS

system. Cloud conditions were identified using visibility and RH sensors integrated into the GCVI system, with a visibility threshold of 1000 m and RH threshold of 95%. The inlet system automatically switched sampling when clouds were detected. During cloud processes, cloud residual and cloud interstitial were alternatively sampled every 30 minutes, with the valve state of 1 for interstitial particles and 2 for cloud residual particles. The setup of sampling system, including all the aerosol measurement instrument, is
given in Fig. 1.

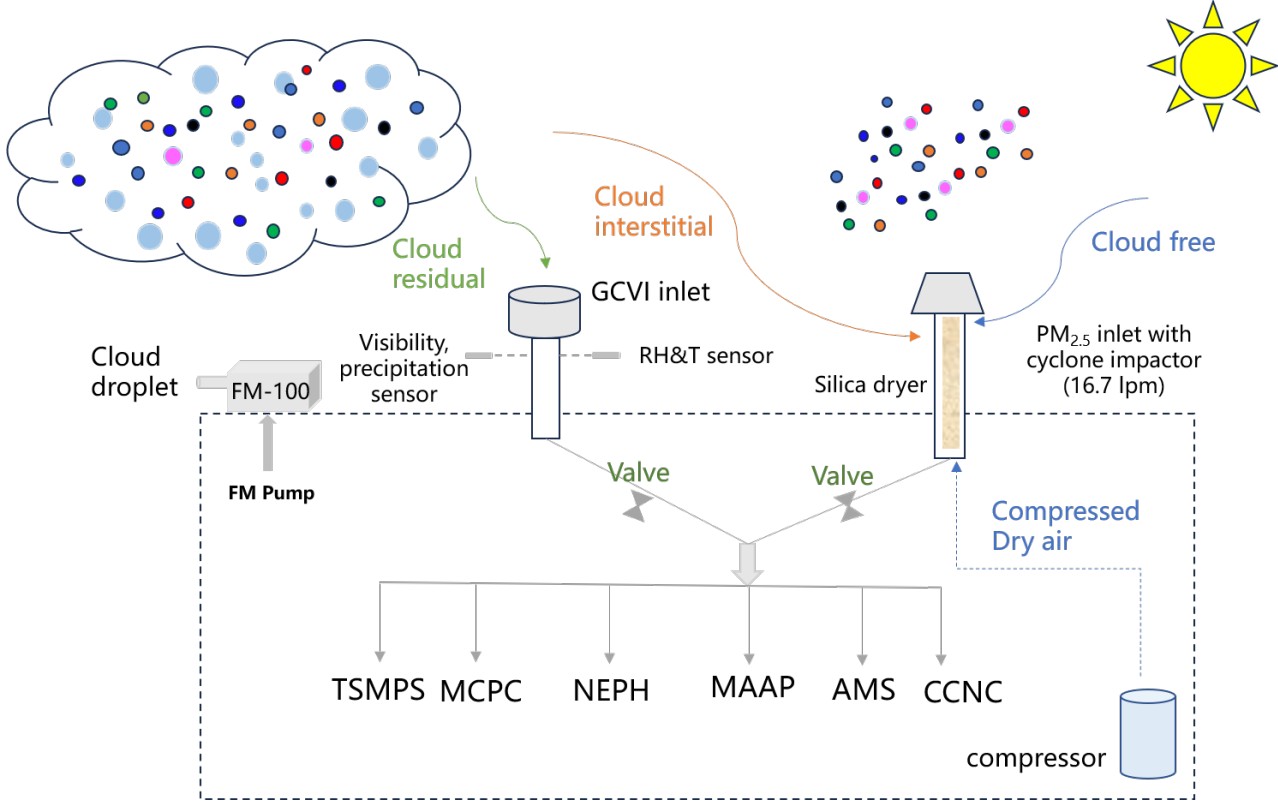

Fig. 1 Setup of the automatic switched inlet system between GCVI inlet and PM$_{2.5}$ inlet, by sampling the cloud free, cloud residual and interstitial particles, including a Twin Scanning Mobility Particle Sizer (TSMPS), Mixing Condensation Particle Counter (MCPC), Aerosol Mass Spectrometer (AMS),
cloud condensation nuclei counter (CCNC), Multi-Angle Absorption Photometer (MAAP), Nephelometer (NEPH), and fog monitor (FM).

The GCVI inlet is capable of capturing cloud droplets with aerodynamic diameters above 7.8 μm by setting the airspeed and counter flow to 90 m s$^{-1}$ and 4 L min$^{-1}$, respectively. The droplets were dried within the GCVI to RH lower than 20% (18.9 ±3.4%) and were subsequently fed into various aerosol
measurement devices. Details of the GCVI system can be found in other studies ( Karlsson et al., 2021). The cut size of GCVI depends on the counterflow and air speed flow, as well as the physical parameters of CVI (Shingler et al., 2012). It is worth noting that the GCVI tends to yield a higher number concentration of cloud particles compared to the actual ambient cloud particle concentration, which

should be corrected for by an enrichment factor (EF). The EF was calculated based on the GCVI sampling flow settings, airspeed within the wind tunnel, and its geometry configuration, as recommended by Shingler et al. (2012). In this work, an EF of 5.9 was derived for airspeed of 90 m s$^{-1}$.

Besides the EF correcting, the GCVI sampling efficiency needs to be considered (Karlsson et al., 2021; Pereira Freitas et al., 2024). The number concentration of cloud residuals can be estimated by integrating the droplets above the cut size of GCVI, from the cloud particle size distribution (as measured by a fog monitor). First, the fog monitor data was corrected based on the equation of $\eta_{tot}(D_{pg}) = \eta_{smp}(D_{pg}) \times \eta_{tsp}(D_{pg})$ (Spiegel et al., 2012), where $\eta_{tot}$, $\eta_{smp}$, $\eta_{tsp}$ is the total counting efficiency, sampling efficiency and transport efficient of the fog droplets with the size of $D_{pg}$. $\eta_{smp}$ of fog particles with $D_{pg}$< 20 μm is approximate 1.0, $\eta_{tsp}$ for particles < 10 μm is about 0.9-1.0 and for particles of 10-20 μm is about 0.85-0.90. In our study, the fog droplets concentrated below 20 μm, with the number concentration accounting for 94±4.5% to the total fog droplets. Thus, we only applied 0.95 for particles < 10 μm and 0.90 for 10-20 μm droplets to correct the transport loss, and 1.0 for sampling loss. With this calculation, total counting efficiency for fog droplet data is 0.95. Second, the MCPC data after GCVI inlet can be corrected by the corrected fog droplet size distribution multiplying by transmission efficiency experimentally determined by Shingler et al. (2012). Finally, the sampling efficiency of GCVI inlet can be derived based on the linear regression between the cloud residual particle number concentration (from MCPC) and the corrected cloud particles. In this work, the linear regression between $N_{mcpc}$ for cloud residual particles with the corrected cloud droplet number concentration above 8 μm from FM was conducted. The slope of the linear regression and the R$^2$ value (coefficient of determination) was 0.68 and 0.69, respectively, as given in SM Fig. S2.

The TSMPS system measured the particle number size distribution (PNSD) within the range 10-850 nm in mobility diameter. And the total particle concentration was measured by MCPC with cut size of 7 nm. The TSMPS operates at a time resolution of 5 mins, while the MCPC operates at a resolution of 1 seconds. By employing this automatic system, which alternates between the GCVI and PM$_{2.5}$ inlets, we can effectively capture the evolution of physical and chemical properties of cloud interstitial and residual particles. The high voltage and size calibration with PSL of nominal diameter 200 nm (203±3 nm, Duke 3000, Thermo Fisher Scientific, USA) was conducted for TSMPS before the experiment, to ensure the size accuracy. The detail information about instrument calibration and data conversion are given in SM.

The chemical composition of non-refractory PM$_1$, including organic components, sulfate, nitrate, ammonium, and chloride, was derived using HR-ToF-AMS with a 1 min resolution, as described by

Canagaratna et al. (2007). This allowed for the determination of particle mass size distributions for organics, sulfate ($SO_4^{2-}$), nitrate ($NO_3^-$) and ammonium ($NH_4^+$) ions. To convert the inorganics salts into their respective forms, e.g., $NH_4NO_3$, $H_2SO_4$, $NH_4HSO_4$ and $(NH_4)_2SO_4$, an ion-pairing scheme was employed, following the methodology outlined by Gysel et al. (2007). The detail information about AMS calibration is given in SM.

The MAAP determined aerosol absorption coefficients directly, which can be converted to mass concentrations of equivalent black carbon (eBC) with an assumed mass absorption efficiency of 6.6 m$^2$ g$^{-1}$ (Petzold and Schönlinner, 2004).

The supersaturation (*SS*) in the CCNc-100 was set to 0.1%, 0.2%, 0.4%, and 0.7%, respectively, with a time interval of 5 minutes for each *SS* during the experiment. The first minute data for each *SS* was removed as the CCNc needs time for *SS* stabilization. Therefore, a complete SS scan cycle lasted ~20 minutes. The supersaturation of CCNc-100 was regularly calibrated with ammonium sulfate particles (Liu et al., 2023).

Additionally, the number size distribution of cloud droplets, ranging from approximately 2 μm up to 50 μm in optical diameter, was measured using a fog monitor (FM-100, DMT Inc., Boulder, CO, USA) at 1 Hz time resolution, with the airspeed of 14.7 m s$^{-1}$. Based on the FM-100, the parameters characterizing cloud droplets, including number concentration ($N_d$), effective diameter ($D_{pe}$) and liquid water content (LWC) can be derived. FM-100 was calibrated with the glass beads of 8, 15, 30 and 50 μm before the start of field campaign, to ensure the accuracy of cloud droplet size measurement.

## 2.3 Methods and calculations

The scavenging efficiency ($\eta$) of particle number concentration in specific size range or mass concentration of chemical composition can be determined by comparing the mean concentration during the last hour before the onset of cloud episode with the mean concentration of cloud interstitial particles during the first hour after cloud begins, according to the method proposed by Noone et al. (1992):

$$\eta = \frac{C_{pre} - C_{ci}}{C_{pre}} \qquad (1)$$

where $C_{pre}$ presents the mean concentration of particles or chemical composition during the last half an hour before the cloud episode, and $C_{ci}$ represents the mean concentration of cloud interstitial particles during the half an hour after the cloud begins in this study. This formula quantifies the percentage change in concentration from before the cloud episode to the initial hour after its onset, providing insight into the efficiency of scavenging by the cloud. The larger value of $\eta$ indicates more efficiently scavenging of

particles by the cloud processes.

The hygroscopicity parameter ($\kappa$) of bulk aerosols can be predicted based on their chemical composition using the Zdanovskii-Stokes-Robinson (ZSR) mixing rule (Petters and Kreidenweis, 2007). The expression for calculating $\kappa$ is given by Equation (2):

$$\kappa_{bulk} = \sum \varepsilon_i \kappa_i \qquad (2)$$

where $\varepsilon_i$ and $\kappa_i$ is the volume fraction and hygroscopicity parameter of each chemical composition, respectively. $\kappa$ of $NH_4NO_3$, $H_2SO_4$, $NH_4HSO_4$ and $(NH_4)_2SO_4$ was 0.67, 0.92, 0.61 and 0.61 (Raymond and Pandis, 2003; Svenningsson et al., 2006). Due to the diversity of organics (different composition, the oxidation state and multifunctional groups, etc.), their $\kappa$ values can vary considerably, typically ranging from 0.03 to 0.3 (Chang et al., 2010; Liu et al., 2021; Zhang et al., 2023). For this study, a median value of 0.1 was selected for organics. When applying the ZSR method to calculate the hygroscopicity of bulk aerosols under the assumption of an externally mixed state, BC is treated as a pure component, with a $\kappa$ value of 0 commonly used (e.g., Pöhlker et al., 2023).

## 2.4 The intercomparison of cloud residual particles

Total particle number concentration ($N_t$) ranging from 10 to 850 nm was determined by integrating the PNSD obtained from TSMPS. This value was compared with the value directly measured by MCPC ($N_{mcpc}$). The dataset was analyzed by excluding periods of precipitation. On April 21[st] and 22[nd], the CPC of TSMPS system was flooded, leading to the exclusion of affected data. In general, the valid PNSD data accounted for approximately 75% of measurement period, with the fraction of cloud-free (CF), cloud interstitial (CI) and cloud residual (CR) data consisting of 78%, 17.5% and 4.5%, respectively. In this study, four typical cloud processes occurring on April 19[th], 28[th], May 5[th], and 8[th], were analyzed, which were selected based on the availability of PNSD and fog data and not exact same as the parallel study by Liu et al., 2025 (preprint in ACP). For CF conditions, six cases of new particle formation (NPF) events were observed.

It was observed that $N_t$ and $N_{mcpc}$ exhibited strong linear correlation, as shown in Fig. 2. However, $N_t$ was found to be approximately 30% higher than $N_{mcpc}$. For CF particles, the slope ($K$) of the linear fitting between $N_t$ and $N_{mcpc}$ was 1.38, whereas the slopes were 1.29 and 1.09 for CI and CR particles, respectively. Part of the bias could be attributed to the systematic uncertainties inherent to each instrument, as well as different particle size that MCPC (7 nm-2.5 μm) and TSMPS (10-850 nm) measured. However, the more important reason is that the $N_t$ values have been corrected to account for diffusion loss dependent

on the particle size, whereas $N_{mcpc}$ was not corrected as it was the total particle concentration without size information. Based on the diffusion loss calculations (Wiedensohler et al., 2012) as given in SM Fig. S3, it was found the diffusion loss of particles below 100 nm could be 15%, which can explain half of the discrepancy between $N_t$ and $N_{mcpc}$. It was also observed that the PNSD of CF and CI particles was dominated by Aitken mode particles, resulting in higher $N_t$ due to diffusion loss correction. In contrast, for CR particles, the accumulation mode particles were predominated, the effect of diffusion loss correction on particle concentration is less. The discrepancy between $N_t$ and $N_{mcpc}$ for CR particles is approximately 9%.

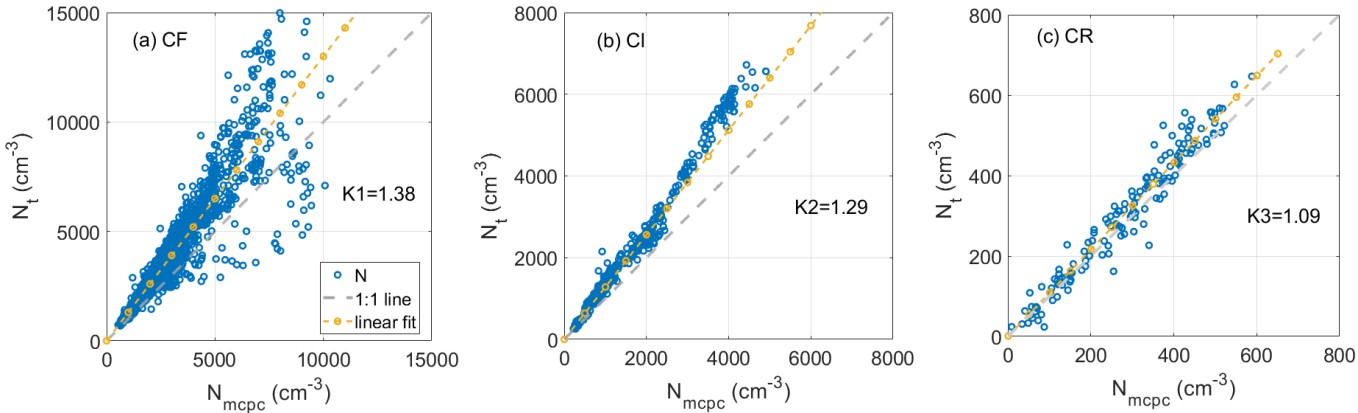

Fig. 2. The scatter plot and linear regression results between particle number concentrations derived from MCPC ($N_{mcpc}$) and TSMPS ($N_t$) for cloud-free particles (a), cloud interstitial particles (b) and cloud residuals (c), respectively. The grey and orange dash lines represent 1:1 line and linear fitting line, and K value is the fitted slope.

## 3. Results and discussions

### 3.1 Overview of PNSDs under different conditions

The mean PNSD data for CF, CI and CR particles, along with the modal fitting results are given in Fig.3, and the statistical values, including the median and standard deviations are for each type particles are given in SM Fig. S4. For CF particles, $N_t$ was $3530 \pm 2113$ cm$^{-3}$, with the mean PNSD being primarily characterized by an Aitken mode, with a geometric mean diameter ($D_{pg}$) of 47 nm and geometric mean number concentration ($N_g$) of 2750 cm$^{-3}$. An accumulation mode was also present, with a $D_{pg}$ of 126 nm and $N_g$ of 1415 cm$^{-3}$. For CI particles, the Aitken mode was the predominant contributor, featuring a $D_{pg}$ of 56 nm and $N_g$ of 1400 cm$^{-3}$. Accumulation mode particles had a minor contribution, suggesting that most of these particles had likely been activated into cloud droplets with diameter larger than 2.5 μm and were unable to pass through the PM$_{2.5}$ cyclone. $N_t$ of the CR particles was 455 cm$^{-3}$, with the observed

particles exclusively within the accumulation mode particles, having a $D_{pg}$ of 220 nm and $N_g$ of 523 cm$^-$ $^3$. Notably, CR particles below 70 nm were absent in the PNSD measurement, which implies that these smaller particles were not either efficiently scavenged by cloud processes or were not capable of being activated into cloud droplets during the cloud processes.

In contrast to polluted regions, a pristine area such as the Zeppelin Observatory near Ny-Ålesund, Svalbard (approximately 480 m above sea level) in the Arctic, exhibits a different PNSD pattern, which is typically dominated by accumulation mode particles peaking round 150 nm which is indicative of cleaner atmospheric conditions. And the mean total particle number concentration of CR particles was approximately 100 cm$^{-3}$ (Karlsson et al., 2021). However, during the summer months, observations sometimes reveal a great contribution from the Aitken mode to the CR particle PNSD. These particles peak at around 60 nm, extending down to the sizes between 20-30 nm (Gramlich et al., 2023; Karlsson et al., 2021). The comparison highlights a shift toward larger particles over 200 nm in size and a higher particle number concentration in polluted regions. Conversely, in the remote Arctic, Aitken mode particles have been found to significantly contribute to the population of CR particles (Karlsson et al., 2022).

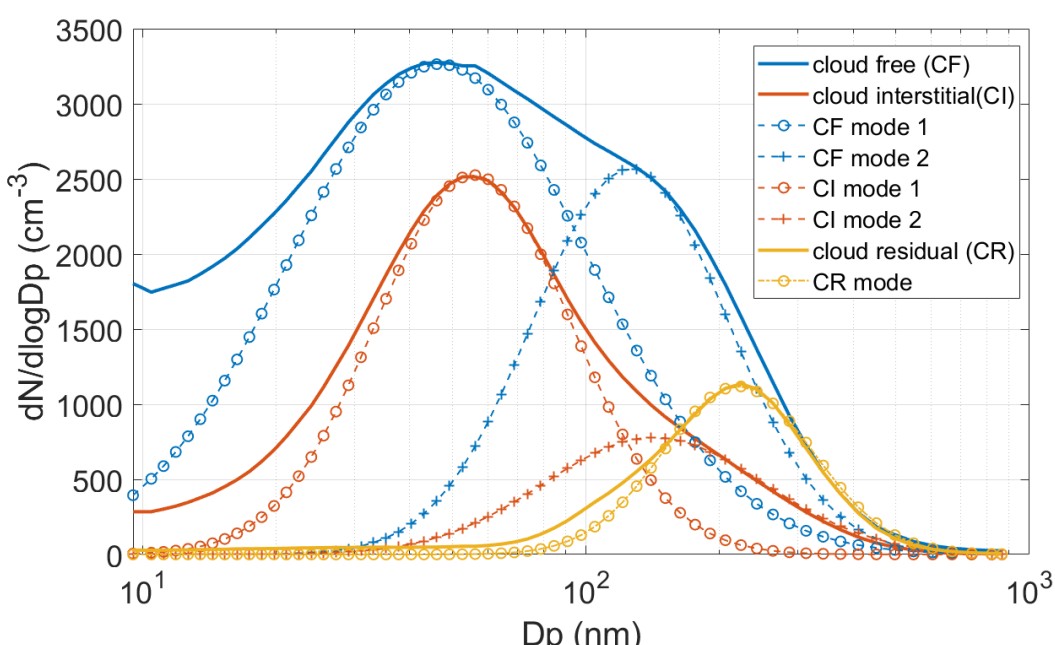

Fig. 3 Mean PNSDs for cloud-free (blue), cloud interstitial (dark orange) and cloud residuals particles (orange) for the entire measurement period, respectively. The dashed lines with circle and cross represent the fitting Aitken and accumulation mode in log-normal scale for each type of PNSD.

## 3.2 Cloud episodes

The evolution of PNSDs during typical cloud episodes was analyzed (Fig. 4), including PNSDs corresponding to different stages: before the cloud presence (30-minute average), cloud presence, cloud development and the mature stage (with 30-minute or 1-hour average) (a-d) and the corresponding size-

resolved scavenging efficiency dependent on time (e-h). The data reveal rapid changes in PNSDs during the cloud process, which typically lasts for 1-2 hours, characterized by a significant reduction in accumulation mode particles. During the mature stage of the cloud episode, occurring 1-2 hours after cloud formation, the PNSDs of CI particles showed only slight changes, particularly for particles larger than 100 nm, which had already been activated into cloud droplets. At this stage, aerosol particles could no longer to be activated into the cloud phase due to the limited availability of water vapor in the ambient environment. For specific cases on April 19 and May 8 (Fig. 4a, d), the number concentration of ultrafine particles with diameter below 100 nm was observed to be higher for CI particles. This suggests that other sources contributed to the elevated ultrafine particle number concentration for these cases, probably related with the changes of air masses, which will be discussed further in the case study below. According to Eq (1), the large value of $\eta$, close to 1, indicating the number concentration of cloud interstitial particles is tending to zero, and all the particles are activated as cloud droplets. Although the PNSDs showed significant variations before and during the cloud process and nucleation scavenging process was time dependent, it was found once the cloud process was observed, the scavenging efficiency tended to 1 for particles above the estimated activation diameter (marked by red box in e-h), indicating these particles were already activated as cloud droplets.

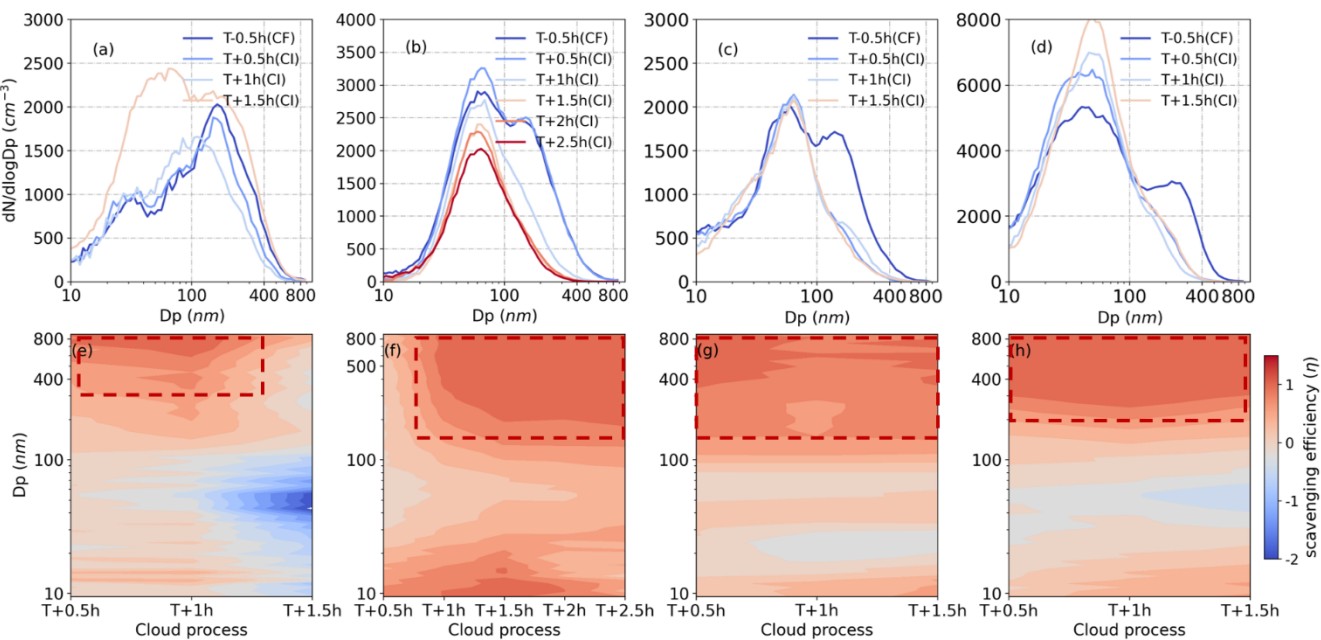

Fig. 4 The evolution of mean PNSDs half hour before the cloud onset and during the cloud processes occurred on April 19, April 28, May 5, and May 8 (a-d), and the corresponding size-resolved scavenging efficiency dependent on time (e-h), respectively. the scavenging efficiency about 0.5 was marked with the red box。

### 3.3 Activation diameter and particle hygroscopicity

Several processes may influence the PNSD of submicron particles at mountain sites, including the entrainment of particles from the free troposphere, NPF events, condensation growth, coagulation, in-cloud scavenging of interstitial particles by droplets, aqueous phase chemistry, advection, activation, wet and dry deposition (Zheng et al., 2018). The evolution of PNSDs on the selected four cloud processes as given in Fig. 4 presented that PNSDs significant changed in the first hour when clouds processed started, with slight changes in the later stage during the cloud. The evolution PNSDs throughout the cloud process enables an approximate estimation of the critical activation diameter. The scavenging efficiency of particles ($\eta_p$) is calculated using the equation (1), according to the changes of particle number concentration of each size bin half hour before (CF particles) and half hour after the cloud processes onset (CR particles). $\eta_p$ values were calculated for the selected four typical cloud processes, including April 19, 28, May 5 and 8, as given in Fig. 5. The $\eta_p$ value of 0.5 indicates that 50% of the particles being activated into CCN, we can deduce that the critical activation diameter ($D_c$) for each cloud process. However, for particles smaller than 50 nm, $\eta_p$ may not accurately represent the activation ratio since these particles tend not to act as CCN. In addition, $\eta_p$ values at some size bins are even negative indicating the particle number concentration increase during the cloud process. As mentioned above, diffusion loss of particles below 30 nm was significant, which also resulted in the measurement uncertainty of PNSD. This observation implies that the evolution in PNSDs before (CF particles) and during the cloud process (CI particles, particularly in the initial stage of cloud formation) can serve as an indicator of the critical diameter for particles activation into cloud droplets under ambient conditions. In support of this method, Hammer et al. (2014) successfully derived the dry activation diameter by examining the difference in PNSDs prior to cloud formation compared to that of interstitial aerosols. However, it's important to note that the evaluation of PNSDs modification during cloud processing did not account for coagulation, evaporation processes and changes in air mass. Thus, it can introduce uncertainty in determining the critical diameter and the CCN number concentration.

$D_c$ of these four cloud processes ranges from 133 nm to 325 nm, which depends on the particle number size distribution, hygroscopicity, mixing state, and the supersaturation of the cloud. The $\kappa$ values were calculated with an hour average before cloud processes, which was 0.22 on April 19, approximately 0.28-0.29 on April 28, May 5 and 8, respectively. On April 19, it showed the largest $D_c$ of 325 nm, corresponding to the weakest hygroscopicity, with highest mass fraction of organics (60-65%) and lowest fraction of nitrate (5-10%). As $D_c$ was normally larger than 130 nm in this study, we calculated the number

concentration of particles larger than 130 nm, marked as $N_{130}$, to represent the potential CCN concentration. Total particle number concentration ($N_t$) and $N_{130}$ was approximately 1500 cm$^{-3}$ and 670 cm$^{-3}$ pre the cloud onset on April 19, which was comparable with the value before the cloud process on May 5, 1700 cm$^{-3}$ and 620 cm$^{-3}$, respectively. However, $D_c$ was smaller on May 5, approximately 133 nm, which probably related with the stronger particle hygroscopicity. $N_t$ and $N_{130}$ was 2400 cm$^{-3}$ and 1000 cm$^{-3}$ pre the cloud process on April 28, which was 5200 cm$^{-3}$ and 1500 cm$^{-3}$ on May 8, indicating a relative polluted condition as influenced by anthropogenic emissions. The corresponding bulk aerosol hygroscopicity was also comparable for the two cases, with the $\kappa$ value of approximately 0.29. However, $D_c$ on May 8 was 199 nm, 25% higher than that on April 28, with $D_c$ of 159 nm. The difference between the PNSDs before the cloud processes could be responsible for the different $D_c$. The lower fraction of $N_{130}$ accounting for $N_t$ indicated that the PNSD was dominated by the particles below 100 nm on May 8. These ultrafine particles can be also hygroscopic, but cannot be activated to be CCN as the supersaturation is probably not high enough. The LWC during the cloud process on May 8 was only 0.025 g m$^{-3}$, which was significantly lower than that on April 28, 0.16 g m$^{-3}$, indicating less available water in the ambient on May 8.

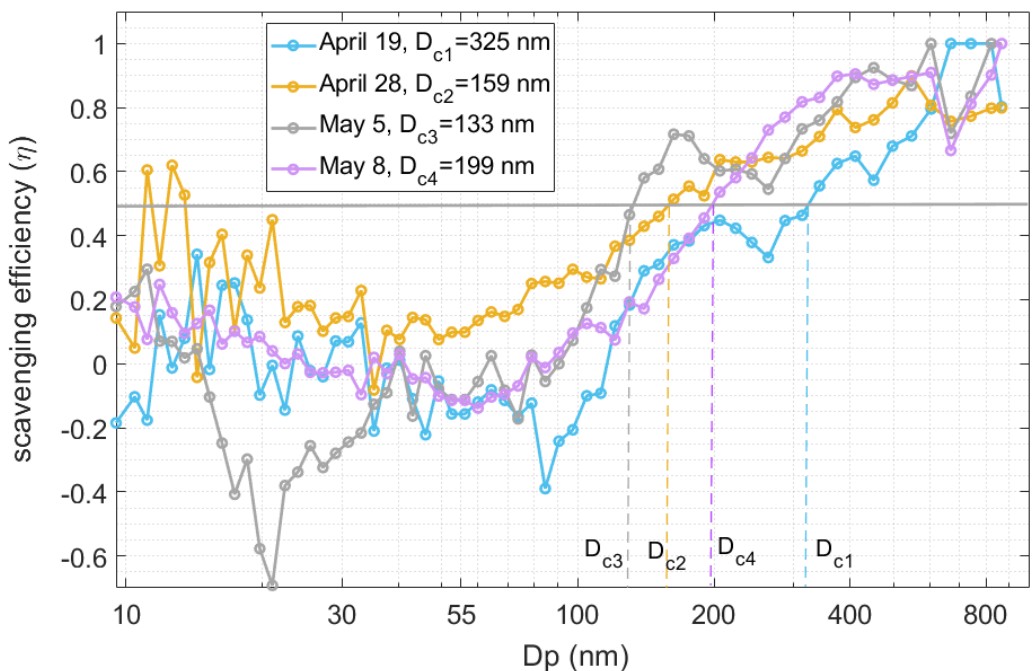

Fig. 5 The size dependent scavenging efficiency ($\eta_p$) for the cloud processes on April 19, 28, May 5 and 8, respectively.

Based on the critical diameters derived from the fast changes of PNSDs before and during cloud formation, the number concentration of CCN can be estimated by integrating from the $D_c$ to 850 nm of the PNSD before the cloud (CF particle from PM$_{2.5}$ inlet), marked as $N_{ce}$. $N_{ce}$ was 132, 648, 423 and 710

cm$^{-3}$ on April 19, 28, May 5 and 8, respectively, and was compared with the measured CCN number concentration ($N_{cm}$) at different supersaturations ($SS_s$), as given in Fig. 6. $N_{ce}$ was generally lower than $N_{cm}$ at $SS$=0.2%, or even lower than that at $SS$=0.1% on April 19. Based on interpolation and extrapolation of $SS$ dependent $N_{cm}$, we can also roughly estimate the corresponding $SS$ in the cloud, which is 0.03%-0.15% for the four selected cloud processes, with the mean value of 0.1%. It was comparable with the estimated $SS$, 0.07±0.02%, based on PNSD and fog microphysics (Liu et al., 2025, preprint ACP). That suggests the determination of $D_c$ and $N_{ce}$ based on the PNSDs before cloud presence is reasonable.

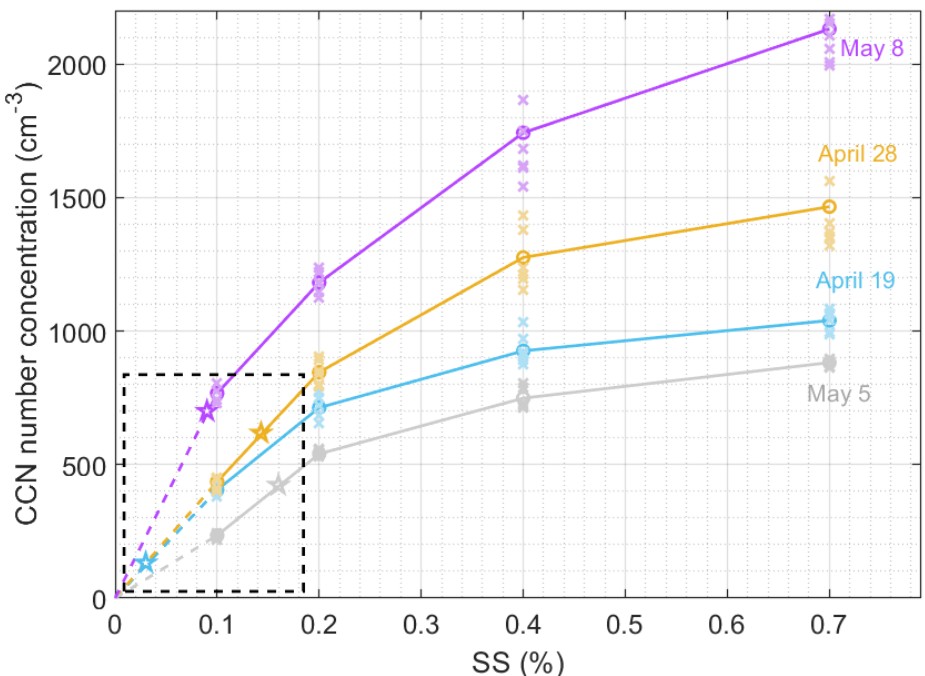

Fig. 6 The measured CCN number concentrations (circle) with different supersaturations and the estimated CCN based on critical diameter (star) on April 19 (blue), 28 (orange), May 5 (grey) and 8 (purple). The cross represents all the measured CCN number concentration in an hour before cloud process and the circles are the mean values

PM$_1$ mass concentration and the mass fractions of each chemical composition are presented in Fig. 7. For CF particles, the average PM$_1$ mass concentration was recorded of 7.3 μg m$^{-3}$. In comparison, PM$_1$ mass concentration for CI and CR particles were lower, which was 3.0 and 4.5 μg m$^{-3}$, respectively. We observed that CI particles had the highest mass fraction of eBC, which resulted in the relatively lower $\kappa$ value as previously discussed. Conversely, a higher nitrate mass fraction in CR particles was associated with an increased $\kappa$ value. Previous study revealed that the formation of sulfate and nitrate within cloud parcels leads to a more internally mixed state of aerosols, and enhance the activation potential of resuspended aerosol particles, thereby influencing their role as CCN, simulating by the particle resolved model PartMC-MOSAIC (Yao et al., 2021). For a more comprehensive understanding of the chemical

species' variability during the cloud process, additional insights will be provided in the forthcoming study by Zhang et al. (currently in preparation). It is important to mention, however, that the chemical composition data for CR particles biased to larger cloud droplets due to the limitations of the GCVI inlet, which samples only cloud droplets larger than 7.8 μm in this study.

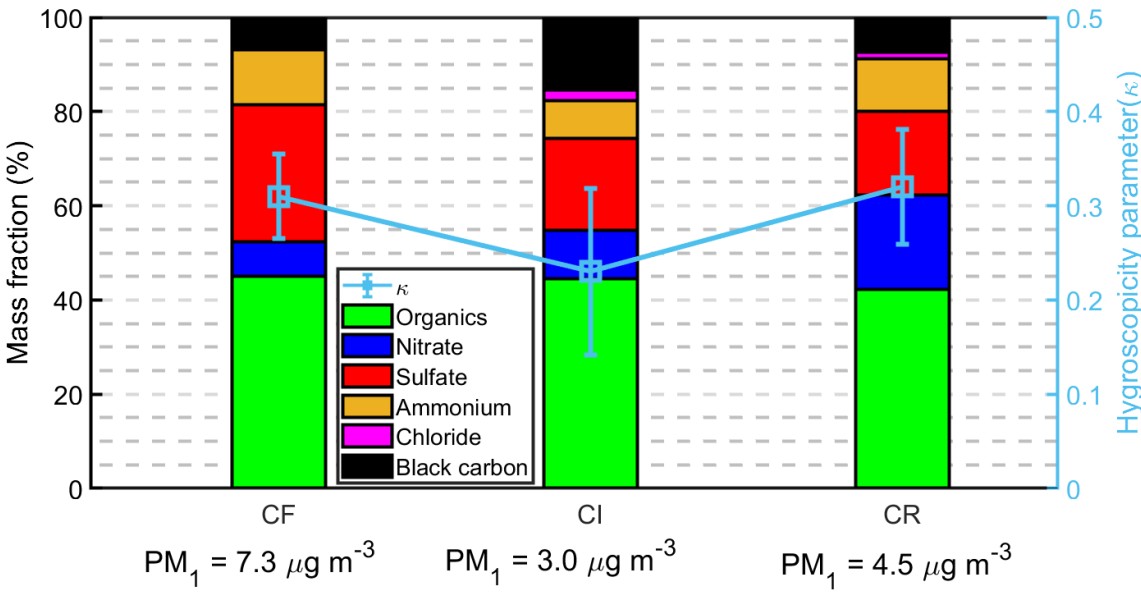

Fig. 7 The mass fraction of organics, nitrate, sulfate, ammonium and chloride for $PM_1$ and mean value (square) standard deviation (error bar) of hygroscopicity parameter ($\kappa$) for cloud free (CF), cloud interstitial (CI) and cloud residual (CR) particles, respectively.

    Collett et al. (2008) have demonstrated that cloud scavenging processes effectively reduce aerosol loadings and alter their hygroscopic properties and size distributions by preferentially removing
hydrophilic species. In our study, CR particles, which represent those particles were activated into droplets in the ambient atmosphere, exhibited the highest mass fraction of nitrate compared to CF and CI particles, correlating with the largest $\kappa$ value observed. When evaluating cloud scavenging efficiency, we compared the removal of the chemical species between CI and CF particles. $\eta_{sul}$ was 0.70, which was significantly higher than that for organics $\eta_{org}$ at 0.62, ammonium ($\eta_{ammo}$) at 0.56, and nitrate ($\eta_{nit}$) at 0.08, indicating
that sulfate aerosols are the most effectively scavenged species. Whereas for the nitrate, the mass concentration didn't show a clear variation, which was 0.67 μg m$^{-3}$ and 0.62 μg m$^{-3}$ for CF and CI particles, respectively, and a little bit higher in CR particle, 1.02 μg m$^{-3}$. That suggests the cloud processes are conductive to the conversion of gaseous $HNO_3$ to particulate nitrate, as both $HNO_3$ and $NH_3$ can be absorbed into cloud droplets (Seinfeld and Pandis, 2016). Further reinforcing the importance of aerosol-
cloud interactions, Ervens et al. (2011) highlighted that cloud processing plays an important role in the formation of both secondary organic and inorganic aerosols. However, the specific mechanisms

underlying these transformations within cloud processes remain unclear and warrant additional research to gain a more comprehensive understanding. For some conditions, for example, on April 19 and May 8, the particle number concentration and mass concentration after the cloud processes increased, depending on the changes of the air masses origin, as will be discussed below.

## 3.4 The microphysics of cloud droplets during a typical cloud process

Precipitation began on the morning of May 6 and continued for almost a full day, ending around 10:00 LT on May 7, during which the observation site remained within the cloud. By 7:00 LT on May 8, visibility increased and RH decreased (Fig. 8). Subsequently, particle concentration began to rise due to the development of boundary layer. During the daytime, visibility gradually decreased, and RH increased to 100%, marking the onset of the cloud process and GCVI system started to work. Concurrently, the number concentration of ultrafine particles also increased, likely influenced by the changing air masses. Six back trajectories were calculated, terminating at observatory station at 2:00, 8:00, 14:00, and 20:00 local time (UTC time+8 h) (in SM, Fig. S5). These trajectories indicate that air masses passed through the North China Plain since from 20:00 LT, transporting significant air pollutants to the mountain site, and the organics and eBC increased gradually since late afternoon (Fig. 8c). The influx of polluted air likely had a substantial impact on the microphysical evolution of the cloud which will be further discussed in the following section.

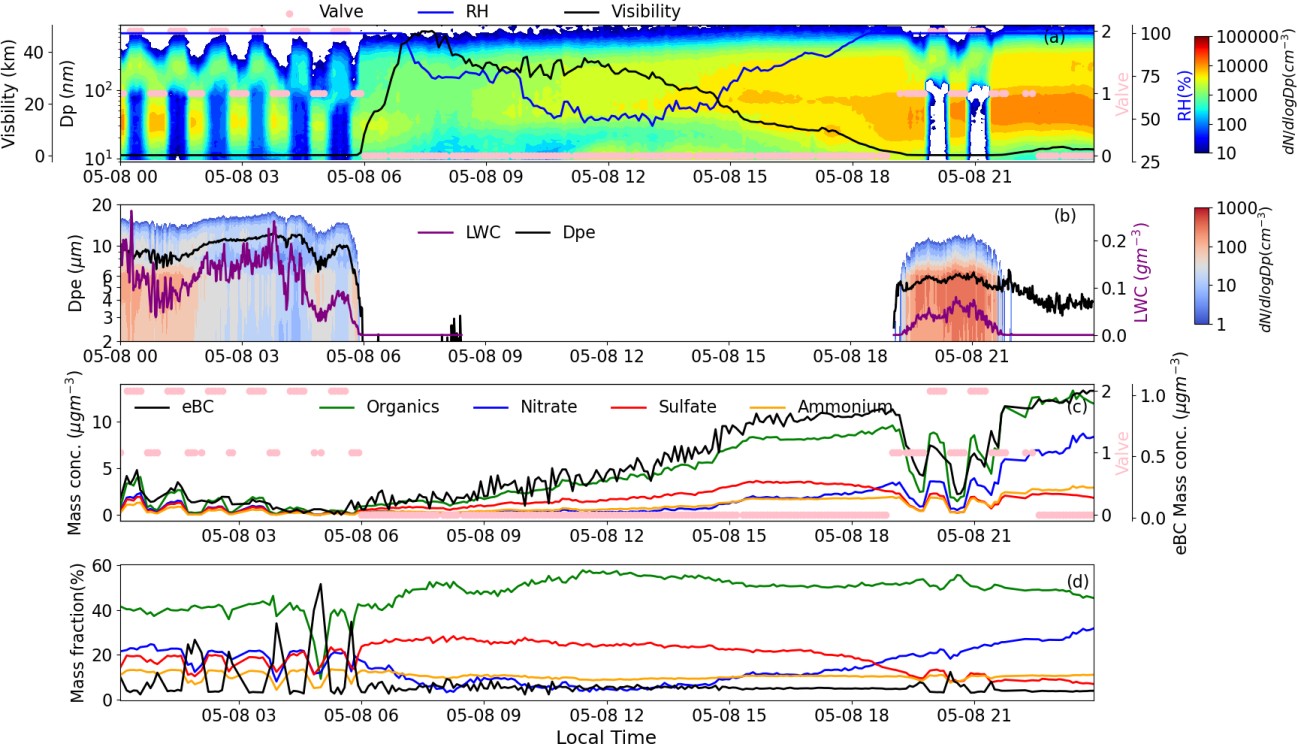

Fig. 8 (a) Particle number size distribution (contour plot), as well as relative humidity (blue line), visibility (black line) and inlet system state (red) on May 8, (b) number size distribution of cloud

droplet, liquid water content (purple line), geometric mean diameter, $D_{pe}$ (black line), (c) mass concentration of chemical composition (organics, nitrate, sulfate, ammonium and equivalent black carbon) and mass fraction of each chemical composition (d).

We found although the mass concentration decreased clearly during the cloud process, the mass fraction of each component showed different variation (Fig. 8d). During the cloud episode of 19:15-22:00 LT, it was found eBC mass fraction decreased for CR particles, while increased for CI particles. That is because eBC is usually hydrophobic, which is difficult to be activated as cloud droplets. However, the mass fraction of sulfate was higher for CR that that for CI particles. It was interesting nitrate mass fraction remained increasing since late afternoon, which needs to be further quantified by the modelling work.

High time resolution measurements of number size distribution of CF, CI and CR particles, and cloud droplets provided detailed insights into the rapid changes in particle concentrations before and after cloud presence (Fig. 9). The evolution of PNSDs of CI and CR particles from 19:10 to 19:40 LT, illustrated by colored lines transitioning from dark blue to dark red (Fig. 9a, b), showed that the presence of cloud passage was observed around 19:15 LT. At this time, the number concentration of particles larger than 200 nm ($N_{200}$) sharply decreased, indicating these particles were activated as cloud droplet. In contrast, the ultrafine particles smaller than 100 nm didn't exhibit significant variation in the cloud interstitial particles, suggesting they were not activated. Correspondingly, the number concentration of cloud droplet increased significantly at 19:15 (Fig. 9c, f), with effective diameter of fog droplet ($D_{pe}$) increasing from 2.5 µm at 19:00 to approximately 5.0 µm at 19:15 LT, and then remaining relatively consistent as cloud droplets formed. Before cloud formation, $N_{200}$ was approximately 550 cm$^{-3}$, and gradually decreased from 19:15 LT to approximately 100 cm$^{-3}$ by 19:40 LT (Fig. 9d). Simultaneously, the number concentration of cloud droplet ($N_d$ in Fig. 9c) and CR ($N_t$, $N_{mcpc}$ in Fig. 9e) increased to approximately 400-500 cm$^{-3}$ during this period.

Hydrated aerosols of the accumulation mode co-existed with droplets, as interstitial non-activated aerosols. Their size continued to increase, and some aerosols achieved diameters larger than 2.5 µm. In the previous study, it has been reported that the mean transition diameter between the aerosol accumulation mode and the small droplet mode was 4.0±1.1 µm (Elias et al., 2015). The contribution of interstitial particles to the light extinction can not be ignored (Liu et al., 2025, preprint in ACP). For the hydrated aerosols larger than 2.5 µm, they will be removed by the impactor and can not enter the inlet system, which resulted in the underestimation of cloud interstitial particles. However, this part of hydrated aerosol can be detected by FM and mistaken as cloud droplets. Unfortunately, it is difficult to differentiate

these aerosols quantitively. But in this case, the small droplet mode around 2.5 μm by FM, also corresponded to the highest concentration of LWC, indicating the contribution by cloud droplets is overwhelming. The parameters including number concentration of CF, CI and CR particles, as well the cloud droplets properties, mass concentration and hygroscopicity parameter for the selected four cloud processes are also supplemented in SM table 1.

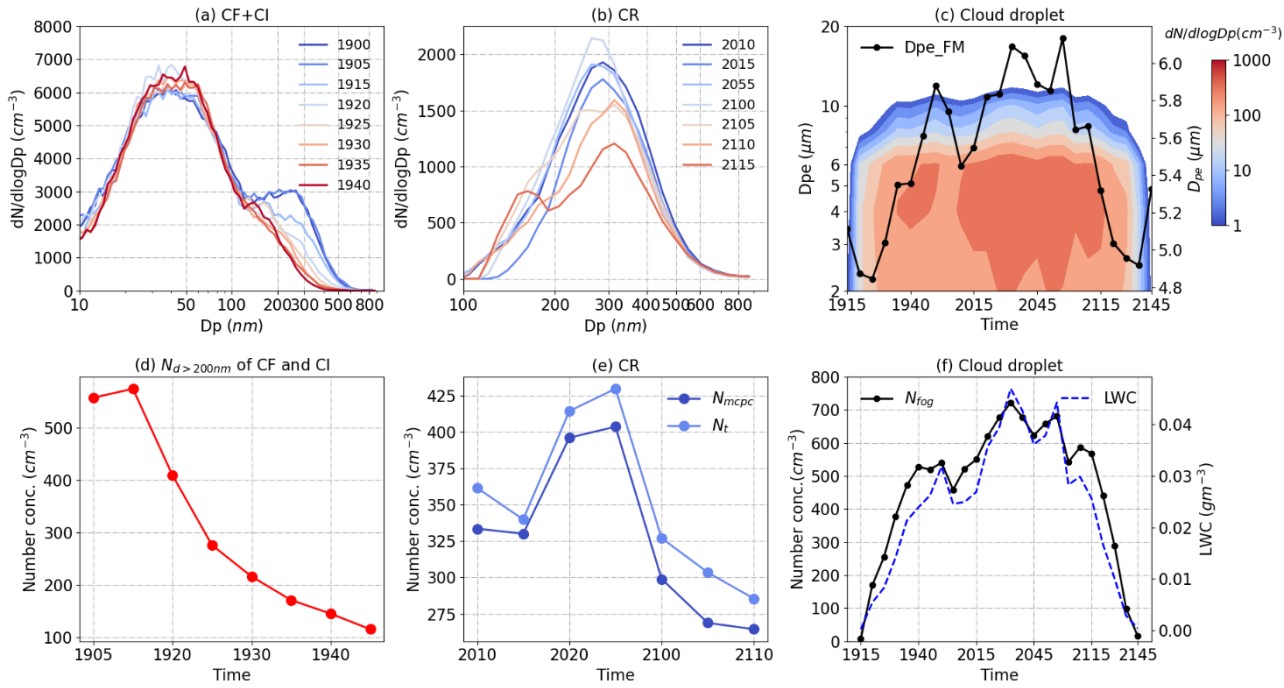

Fig. 9 Number size distribution of cloud free (19:00 and 19:05 LT) and cloud interstitial particles (19:15-40) from PM$_{2.5}$ inlet (a), and cloud residual particles from 20:10-21:15 from GCVI inlet on May 8 (b), the number size distribution of cloud droplets with geometric mean diameter ($D_{pg}$) (c), and integrated number concentration of particles above 200 nm ($N_{200>200\ nm}$, d) and cloud residual particles from TSMPS ($N_t$) and from MCPC ($N_{mcpc}$) (e) corresponding to the time period of (a) and (b), respectively, and the number concentration of cloud droplets and liquid water content (f).

The influence of submicron particles on cloud droplets was analyzed using scatter plots of cloud droplet parameters (LWC, $N_d$, and $D_{pe}$) derived from the FM-100, with the hygroscopicity ($\kappa$), geometric mean diameter ($D_{pg}$), number concentration from MCPC ($N_{mcpc}$) for cloud interstitial (CI) and residual (CR) conditions, respectively, on May 8 case (Fig. 10). For CI particles, stronger hygroscopicity was found with higher number concentration and larger $D_{pg}$ (Fig. 10a). It has also been reported that during the aging process of particles at elevated concentrations, their hygroscopicity can increase (Zhang et al., 2023). However, $N_{mcpc}$ showed a negative relationship with $\kappa$ for CR particles (Fig. 10b). Particles with stronger hygroscopicity were found to relate with lower $N_d$, LWC and smaller $D_{pe}$, as illustrated in Fig 10c, d. This

suggests that aerosols with enhanced water uptake abilities compete with the fog droplets to absorb liquid water from the ambient environment, thereby hindering further growth of cloud droplets. This suggests that numerous aerosol particles in the ambient atmosphere are sufficient to uptake water vapor, thereby limiting supersaturation levels. Previous studies have shown elevated particle number concentration can result in lower supersaturation levels during cloud formation. This finding is consistent with research conducted at a suburban site in Paris, which reported similar results (Bott, 1991; Mazoyer et al., 2019).

The study of hygroscopicity, particle number size distribution on the cloud microphysical properties was also conducted for all the cloud processes during the campaign, however, the overall correlation was weak (Fig. S6 in SM). That indicated the complex evolution of cloud microphysical properties during the cloud life cycle does not exhibit a consistent trend with particle hygroscopicity (composition) and size. Indeed, cloud microphysics vary greatly from case to case and depend on numerous factors. This complexity underscores the need for comprehensive and long-term observations to better understand cloud microphysics and its interactions with aerosol particles. Furthermore, only four typical cloud processes were analyzed using the limited dataset, which introduced uncertainty in the data analysis.

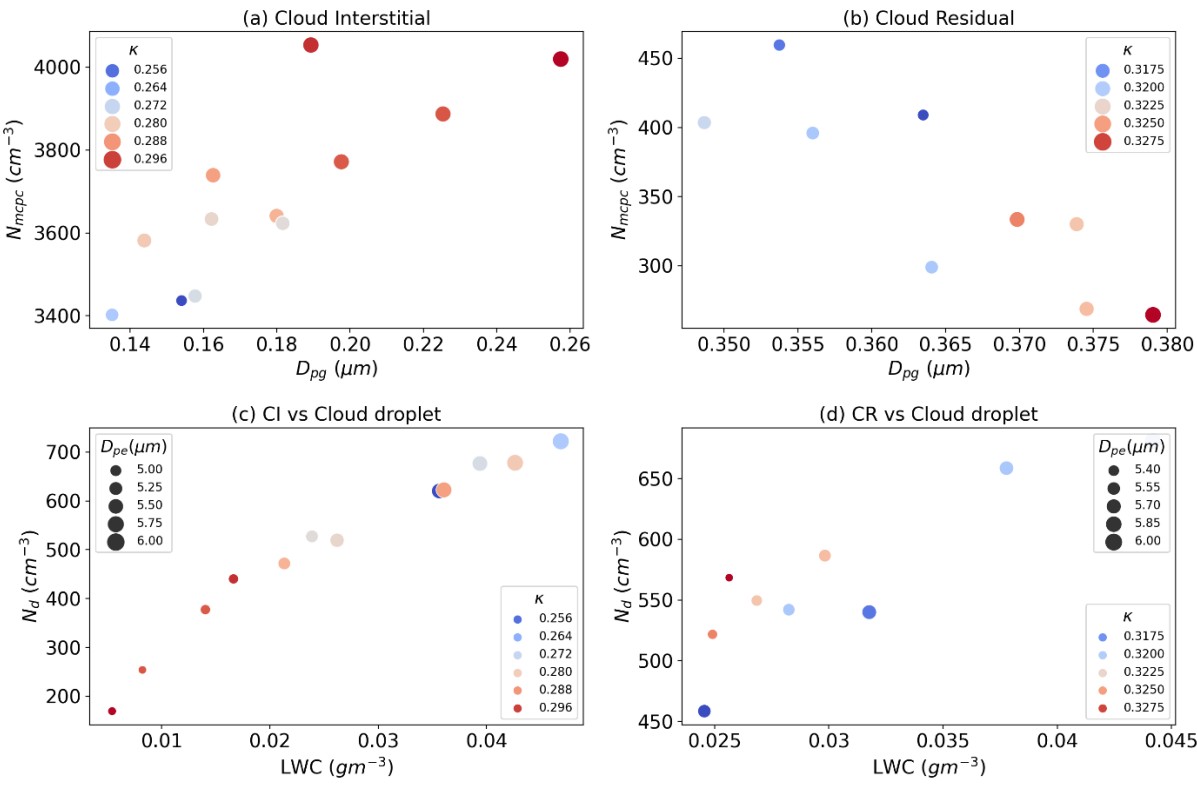

Fig. 10 The scatter plots of particle hygroscopic parameter ($\kappa$), geometric mean diameter ($D_{pg}$), number concentration from MCPC ($N_{mcpc}$) for cloud interstitial, CI (a) and cloud residual (CR) particles (b), as well the scatter plots of $\kappa$ and cloud droplet parameters (LWC, $N_d$, and $D_{pe}$) for CI (c) and CR (d) on May 8 case, respectively.

## 4.  Conclusions

This study employed an automatic inlet system, alternating between GCVI and PM$_{2.5}$ cyclone, to couple with various aerosol measurements, enabling the derivation of the particle number size distributions (PNSDs), chemical composition for cloud free (CF), interstitial (CI) and residual (CR) particles at a mountain-top station in the Yangtze River Delta region of China during the spring season. The findings revealed significantly modifications to PNSDs induced by cloud processes. Before the presence of cloud, PNSDs of CF particles were characterized by a major Aitken mode and a secondary accumulation mode. In contrast, CI and CR particles exhibited unimodal distributions, with Aitken and accumulation modes dominating, respectively. The study identified four typical cloud processes with visibility below 1000 m, without precipitation influence during the measurement. Comparison of PNSDs before and after the onset of cloud processes revealed rapid changes occurring within minutes. Particles larger than approximately 100-200 nm were observed to be activated as cloud droplets and subsequently removed from the aerosol system. In some cloud cases, the number concentration of Aitken mode particles was 50% or even higher for CI particles compared to CF particles before the onset of cloud episode as influenced by the changes of air masses. The critical diameters required for activation as cloud droplets were estimated to be approximately 133-325 nm, depending on the particle hygroscopicity and number size distribution for each individual case. As compared with the measured CCN concentration with different supersaturations ($SS_s$), the estimated $SS$ under the ambient was normally below 0.1% in this study.

For a typical cloud process occurring on May 8, the decreasing trend of CI particles larger than the critical diameter, approximately 200 nm for this case, agreed well with the increase trend of cloud droplets number concentration derived by fog monitor. On May 8 case, it was observed that particles with stronger hygroscropicty and larger size in the CI and CR types were typically associated with lower number concentrations and smaller effective sizes of cloud droplets. This suggests that aerosols with enhanced water uptake abilities compete with the cloud droplets to absorb liquid water from the ambient envrimnent, thereby hindering further growth of cloud droplets. However, due to the limited dataset of cloud events, the mechanisms of removal and chemical transformation in cloud processes remain unclear and more experimental data is needed to support the further research.

## Acknowledgments

This research was supported by supported by the National Key R&D Program of China (grant no. 2023YFC3706305), National Natural Science Foundation of China (42275121, 42475121), China Meteorological Administration Innovation and Development Project (CXFZ2024J039), Chinese Academy of Meteorological Sciences (2023Z012, 2024Z006), and Innovation Team for Haze-fog Observation and Forecasts of MOST.

## Author contributions

XS conducted the measurements, analyzed the data, and wrote the original draft. QL and JS designed the experiment, QL, BQ, YZ, QM, XH, JL, SL, AY, and LL contributed to the measurements, instrument maintenance, data analysis, and results discussions. JS, HC, and XZ reviewed and finalized the article. XS, JS, and QL contributed to fund acquisition.

## Competing interests

The authors declare no conflict of interest.

## Data availability

The data in this study are available at https://doi.org/10.5281/zenodo.13918793. (Shen et al., 2024).

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
