# Peer review of "Measurement report: The influence of particle number size distribution and hygroscopicity on the microphysical properties of cloud droplets at a mountain site"

_EGUsphere, 2024_

## Author Comment (AC2)

We would like to thank the reviewers for the thoughtful and insightful comments for this measurement report. The reviewer provided valuable insights on the use of GCVI and data processing methods, which helped us improve the manuscript and will be beneficial for our future research on aerosol-cloud field campaign and research. The manuscript has been revised accordingly, and our point-by-point responses are provided in below and given in blue font.

The work by Shen et al. addresses the influence of aerosol particles on cloud microphysics, focusing on aerosol size distributions and hygroscopicity at a mountain site in the Yangtze River Delta, China. While the topic is scientifically significant, the manuscript falls short of the required standards in several critical areas. Crucially, it omits essential details about the experimental setup and analysis, such as the GCVI system's performance evaluation, operation and calibration of key instruments, and clear descriptions of data processing methods. These gaps hinder the reproducibility of the research and cast doubt on the reliability of the results, particularly those derived from GCVI measurements. The limited discussion of uncertainties further undermines the credibility of the findings, as robust conclusions require transparency in assessing potential errors.

Response: We are very appreciated for the reviewer's specific and meaningful comments; the authors have tried the best to improve the manuscript. We provided the detail information about instrument calibration, data validation and correction, and also the data for supporting the conclusions.

Additionally, the manuscript requires significant revisions to improve its structure, language, and clarity. Currently, the text suffers from frequent grammatical issues and disorganized presentation, making it difficult to

follow the authors' arguments. Beyond these editorial concerns, there are substantial scientific gaps, including the insufficient contextualization of findings within existing literature and the absence of a detailed comparison of GCVI-derived results to other methods. Furthermore, the overlap in text and content with a parallel manuscript by Liu et al. (https://doi.org/10.5194/egusphere-2024-2264) raises concerns about potential self-plagiarism and the novelty of the work presented here.

Response: Although both articles present the results of the same observational experiment, they focus on different aspects. Liu et al., 2024 highlighted how the cloud interstitial and cloud droplets influenced the visibility and proposed a visibility parameterization scheme. And this manuscript focused on the how the particle number size distribution, chemical composition changes and the relationship between cloud droplets. More explanation is given in below and we have revised the introduction of the observational methods to avoid the repetition.

Overall, the manuscript requires major revisions in both scientific content and editorial quality to meet the high standards of ACP. These revisions are essential not only to enhance clarity and rigor but also to ensure the integrity and reproducibility of the research. Extensive comments are provided below to assist the authors in addressing these issues.

**Detailed comments (in order of appearance):**

- Line 34: This sentence needs to be revised. Aerosols don't "decrease and increase rainfall as a result of their radiative forcing"

They can also decrease and increase rainfall as a result of their radiative forcing and CCN properties suppress precipitation as the amount of solar radiation reaching the land surface was decreased, and enhance precipitation by accelerating the conversion of cloud water by cloud seeding (Rosenfeld et al., 2008).

- Line 38: Reference(s) missing.

The reference has been supplemented. "Pöschl, U., 2005, Atmospheric Aerosols: Composition, Transformation, Climate and Health Effects, Angew. Chem. Int. Ed., 44, 7520-7540. "

- Line 39-40: What kind of scavenging do you mean here? Nucleation of impaction scavenging? I think you need to make this distinction already clear earlier on. Also, do these values relate to fog or clouds?

It has been revised to "In the cloud passage investigated through airborne measurements, it has been found the nitrate entered the cloud droplets and governed by the gas-phase mass transfer process, whereas much of the sulfate in the cloud water is the result of nucleation scavenging (Hayden et al., 2008). In the in-situ measurement of fog events in Po Valley, Italy, the nucleation scavenging efficiency of inorganics species was 60-70%, and 40-50% for organics and black carbon (Gilardoni et al., 2014). Although the fog scavenging processes include impaction and nucleation scavenging, generally, nucleation scavenging process is dominant (Seinfeld and Pandis, 2016)"

- Line 44-45: Cloud particles can also capture non-soluble particles. Revise and add reference(s).

Cloud droplets can capture soluble particles (e. g. sulfate, salts), non-soluble particles (e. g. black carbon, biological aerosol particles), gases through nucleation and impaction processes (Pereira Freitas et al., 2024; Zieger et al., 2023), and other cloud droplets through collisions and coalescence (Wood, 2006).

- Line 46-49: Not the precipitation evaporates but the cloud droplets/particles. The aerosol, which may also consist of soluble material, is not "reemitted into the atmosphere". They are also there! Please revise.

However, if the precipitation does not occur and cloud droplets evaporates instead, dissolved materials and some insoluble components in residual particles can modify the particle mixing states, sizes and even the chemical compositions (Hoose et al., 2008).

- Line 50: Not so many studies have used the ground-based version of the CVI so far. I would suggest revising this sentence. A few more relevant publications could be added to this paragraph, discussing the mixing-state, hygroscopicity and coarse-mode/bioaerosols within cloud residuals using the same GCVI as used here (Adachi et al. 2022, Duplessis et al. 2024, Zieger et al. 2023, Pereira Freitas et al. 2024).

Additionally, the effects of cloud processes on particle mixing-state, hygroscopicity, black carbon, and coarse-mode/bioaerosols were also discussed by using the same GCVI in Zepplin (Adachi et al. 2022, Duplessis et al. 2024, Zieger et al. 2023, Pereira Freitas et al. 2024).

- Line 57-59: Why is the nonlinearity between cloud droplet number and aerosols (I assume concentration?) a challenge? Please refine this sentence.

Satellite observations have revealed a non-linear relationship between cloud droplet number concentration (Nd) and fine mode aerosols concentration (as indicated by Aerosol Index), as the Nd sensitivity decreases with the high aerosol loading, leading to a challenge in accurately assessing cloud droplet changes influenced by aerosols (Jia and Quaas, 2023)

- Line 59: "To enhance the estimation of aerosol-cloud interactions". What kind of estimation? Do you mean to increase our knowledge or process-understanding? Please revise.

To improve the understanding of aerosol-cloud interactions through both observational and modelling approaches, in-situ measurements of aerosol and cloud variables at mountain sites are particularly helpful and essential.

- Sect. 2.1: Are there any existing publications describing the aerosol and meteorological conditions at the site? If so, maybe reference them here.

There has not any publication about this observatory. Another paper drafted by Liu et al., 2024 was just accepted and final version is not published yet.

- Line 96: Maybe also state the mean and STD relative humidity here.

During the measurement, the mean and standard deviation value of temperature and relative humidity (RH) was $12.4\pm2.7$ $^{o}$C and $88.2\pm15.8\%$, respectively, with 14 days with precipitation. For $PM_{2.5}$ inlet system, the aerosol humidity was $24.3\pm6.4\%$ after passing through the dryer system.

- Line 93-95 and 101-102: This is not really relevant information. The LabView programme information and the definition of the values given for the status of the valve would ideally be moved to the data description (e.g. within a read-me file).

This information has removed.

- Figure 1:
    - I assume that the colored spheres within the cloud should represent the interstitial aerosol, while the blue dots are the cloud droplets. The interstitial aerosol should also be shown outside the cloud, as they will be sampled during cloud free conditions as well. Ideally they will be smaller compared to the droplets.
    - Where was the RH measured?
    - The PM2.5 inlet is not properly described. Which total flow was applied? How was it measured/ensured? What kind of cyclone/brand was used?

- The position of the fog monitor is a bit surprising. Were cloud droplets really measured at the site or on the roof? A photo of the set-up within the SI might be useful.

- The GCVI inlet also includes the visibility and precipitation sensor, as well as the small weather station. Please add it to the graph.

- CCN should be CCNC.

The figure has been modified according to the reviewer's comments. Some other aerosol instruments used in this campaign was also supplemented, although the date was not discussed in this manuscript yet. In the schematic diagram, aerosols are represented by different colors, including nitrate (blue), sulfate (red), organics (green), ammonium (orange), black carbon (black), et. The cloud droplets are represented by light blue. A photo of the experimental set-up and station location is also given in the supplementary material. Aerosols were sampled through a $PM_{10}$ impactor and a $PM_{2.5}$ cyclone (16.7 L min$^{-1}$, Thermo Fisher Scientific, USA) in sequence, and the actual total flow rate through the sample inlet was 14.5 lpm, which has been added in the manuscript.

[Figure]

Fig. s2 Setup of the automatic switched inlet system between GCVI inlet and PM$_{2.5}$ inlet system.

[Figure]

Fig. s3 The photo of the experimental set-up and station location

- Line 110-111: Please properly describe what is the air speed (measured within the wind tunnel) and what are the set sampling and counter flows. What was the total instrument sampling flow?

The setting the airspeed and counter flow to 90 m s$^{-1}$ and 4 lpm, respectively. The total flow rate of PM$_{2.5}$ inlet system was 14.5 lpm, including TSMPS (3.5 lpm), Nephelometer and MAAP (9.0 lpm, they are connected in series), MCPC (1.0 lpm), AMS (0.5 lpm), CCNC (0.5 lpm).

- Line 112: Please also state the mean and standard deviation of the sampling RH.

After GCVI inlet, the cloud residual particles were measured by TSMPS and the aerosol RH was also recorded by the RH sensor, which was 18.9 ±3.4%. So we revised this sentence to "The droplets were dried within the

GCVI to RH lower than 20% (18.9 ±3.4%) and were subsequently fed into various aerosol measurement devices."

- Line 113: The referencing regarding the GCVI is not correct here. Roth et al is a different CVI system and Shingler et al. described the CVI used within the GCVI (and not the wind tunnel, etc). A proper evaluation of the GCVI system has so far only been done in Karlsson et al. (2021).

Thanks for the comments. The reference has been corrected here.

- Line 115: The GCVI does not "tend" to yield a higher number concentration. The enrichment or enhancement factor is the result of the aerosol concentration being concentrated in the CVI inlet. The authors completely miss the evaluation of the sampling efficiency of the GCVI, which is also dependent on the ambient cloud droplet distribution. For this, it is needed to compare the cloud residual number concentrations with the parallel measured droplet size distributions of the fog monitor (see Karlsson et al., 2021). Alternatively, the sampling efficiency can also be determined by comparing the accumulation or coarse mode number concentrations of the cloud residuals to the ambient aerosol measurements (see Karlsson et al., 2021 and Pereira Freitas et al., 2024).

Thanks for the comments and we have corrected the GCVI sampling efficiency according to Karlsson et al., (2021), Pereira Freitas et al., (2024), Spiegel et al., 2012, and Shingler et al. (2012) as recommended. The slope of the linear regression (slope) and the $R^2$ value (coefficient of determination) was 0.68 and 0.69, respectively. First, the fog monitor data was corrected based on the equation of $\eta_{tot}(D_{pg}) = \eta_{smp}(D_{pg}) \times \eta_{tsp}(D_{pg})$ (Spiegel et al., 2012), where $\eta_{tot}$, $\eta_{smp}$, $\eta_{tsp}$ is the total counting efficiency, sampling efficiency and transport efficient of the fog droplets with the size of $D_{pg}$. $\eta_{smp}$ of fog particles with $D_{pg} < 20$ μm is approximate 1.0, $\eta_{tsp}$ for particles $< 10$ μm is about 0.9-1.0 and for

particles of 10-20 μm is about 0.85-0.90. In our study, the fog droplets concentrated below 20 μm, with the number concentration accounting for 94±4.5% to the total fog droplets. Thus, we only applied 0.95 for particles < 10 μm and 0.90 for 10-20 μm droplets to correct the transport loss, and 1.0 for sampling loss. With this calculation, total counting efficiency for fog droplet data is 0.95. Second, the MCPC data after GCVI inlet can be corrected by the corrected fog droplet size distribution multiplying by transmission efficiency experimentally determined by Shingler et al. (2012). Finally, the sampling efficiency of GCVI inlet can be derived based on the linear regression between the cloud residual particle number concentration (MCPC data) and the corrected cloud particles. In this work, the sampling efficiency was 0.68, and all the cloud residual particle number concentration and mass concentration have been revised. And the details of data correction have been supplemented in the supplementary material.

[Figure]

Fig. s4 The scatter plot of number concentration of cloud residual particle from MCPC of GCVI inlet and the corrected fog monitor data with GCVI sampling efficiency.

- Within the method section, the authors use exactly the same sentences as in their parallel submitted manuscript by Liu et al, currently in discussion (https://egusphere.copernicus.org/preprints/2024/egusphere-2024-2264/). This can be regarded as self-plagiarism. It is also striking that the same four days from April/May 2023 are used in

both manuscripts. It almost feels that this work here could be combined with the manuscript by Lui et al.

This manuscript (Shen et al., 2024) and Liu et al., 2024 are both the results of the same campaign, with the same inlet and instruments, but focused on different topics. Liu et al., highlights how the fog droplets and interstitial particles influence the low visibility during cloud episodes and Shen et al., focused on how the particle size changed during the cloud process and submicron particles size and hygroscopicity influence the cloud droplets. The cloud/fog events listed in these two manuscripts have some overlapping as they describe the same measurement. The identification of cloud/fog events based on the same principle (with thresholds set at 1000 m for visibility and 95% for RH), however, it also depends on the available data used for discussion. Liu et al., listed the case of April 11-12, 21, 28, May 6 and 8, whereas in Shen et al., cloud case of April 19, 28, May 5 and 8. In Liu et al., the case of April 11-12 was discussed in detail and in Shen et al., we focused on May 8 as pollution occur before cloud episode, which facilitated the study on the influence of high loading particles on cloud process. As there are some repetitive descriptions about experiments and instruments, we also revised the sentences accordingly.

- Section 2.2: Basic information like sampling flows, applied corrections and performed calibrations are missing. This is important information which needs to be added for all instruments! A few important points:
  - How was the TSMPS calibrated? Was an impactor installed? Flows? How do integrated and total number concentration compare? Are the size distributions corrected for difusion/impaction losses? Details on the inversion and multiple-charge correction?

About the TSMPS (TROPOS, Germany) applied in this work, we followed the calibration and data inversion routine as recommended by Wiedensohler et al., (2012). The size (PSL of 200 nm), sample flow and high voltage calibration was conducted before the start of this field

campaign. The tube length and flow rate were also recorded to correct the diffusion loss. The multiple-charge correction and diffusion loss have been conducted by a custom-made data inversion software to make sure the accuracy of PNSD data. The above information have been supplemented in the manuscript.

Reference: Wiedensohler, A., Birmili, W., Nowak, A., et al., Mobility particle size spectrometers: harmonization of technical standards and data structure to facilitate high quality long-term observations of atmospheric particle number size distributions, Atmos. Meas. Tech., 5, 657‒685, https://doi.org/10.5194/amt-5-657-2012, 2012.

- o The MAAP usually runs with a flow of 16.6 lpm. How was it modified to sample behind the GCVI?

According to the manual, MAAP can be operated at the flowrate of 0.5-1.4 $m^3/h$, corresponding to the flowrate of approximately 8.0-23 lpm. In this work, MAAP and Nephelometer are connected in series, with the setting flow rate of 9 lpm. Together with the other instruments, the total flow rate was 14.5 lpm.

- o What kind of corrections were performed for the AMS data? Air beam correction? Correction for the fragmentation of CHO? Did you determine the composition dependent collection efficiency for the AMS?

The calibrations of ionization efficiency (IE) were performed, using size-selected (300 nm) ammonium nitrate particles before and after the experiment. Default relative IE values were used for organics (1.4), nitrate (1.1), sulfate (1.2), ammonium (4.0), and chloride (1.3), respectively. The HR-ToF-AMS collection efficiency (CE) accounts for the incomplete detection of aerosol species owing to particle bounce at the vaporizer, and/or the partial transmission of particles by the lens (Canagaratna et al., 2007). In this study, a composition-dependent CE correction was used, following the methodology described by Middlebrook et al. (2012). Positive matrix factorization (PMF) (Ulbrich et al., 2009) and a multilinear

engine (ME-2) (Canonaco et al., 2013) modelling of high time resolution organic mass spectrometric data from HR-ToF-AMS have also been used to resolve organics into primary organic aerosols (POA) and oxygenated organic aerosols (OOAs), which correspond to different sources and processes (Zhang et al., 2022).

- o The supersaturation schedule of the CCNC is quite short. Could you provide an example time series showing that the time was sufficient to get stable concentrations (especially when changing form 0.7% back to 0.1% SS)?

In this study, the CCN counter was sequentially set to four supersaturation (SS) values: 0.1%, 0.2%, 0.4%, and 0.7%, each for a duration of 5 minutes. The four SS setpoints were sequentially scanned from low to high and then back from high to low to avoid large change of SS in the CCNc column. Due to the cloud chamber inside the CCN counter requires time to stabilize the temperature after each change in *SS*, data measured in the first two minutes of each *SS* were excluded. However, when *SS* goes down from 0.7% to 0.4%, the concentration has large bias, indicating longer time are need for *SS* stabilization and the unreasonable data should be also removed, as the example case of May. 8 shown in Fig s5. We also found the switch between GCVI and PM$_{2.5}$ inlet also influenced the CCNC measurement, and CCN corresponding to the switch time are also removed to minimize the uncertainty of CCN, as shown in Fig s6. In the future field campaign, we need to extend the time of each *SS* scan.

[Figure]

Fig. s5 time series of CCN concentration with different super saturations (ss) on 28, May 8.

[Figure]

Fig.s6 time series of CCN concentration with different super saturations (ss) on April 19, 28, May 5 and 8, respectively.

- o Did you apply any loss corrections for the FM-100? See Spiegel et al. (2012). Did you calibrate it with glass beads?

FM-100 was calibrated with the glass beads of 8, 15, 30 and 50 μm and the date was further corrected by considering the sampling and transport efficiency as above. In this study, the total sampling efficiency of FM was about 0.95, as most droplet particles concentrated below 20 μm.

o In your Fig.1 it seems that all instruments sample behind the GCVI during cloudy conditions (or when the GCVI is on). Could you confirm?

All the instruments, except FM-100, are connected behind GCVI and PM$_{2.5}$ inlet system by a three-valve tube controlled by software automatically. Under cloud free conditions, the three-valve switched to PM$_{2.5}$ inlet. When cloud episode is detected, GCVI starts working and the three-valve switched to GCVI and PM$_{2.5}$ alternatively, with 30 min time resolution. When the three-value module is turned off, it switched to the PM$_{2.5}$ inlet as a default condition. This information has been supplemented in the manuscript.

- Line 139: BC should be called eBC (equivalent black carbon)

It has been corrected.

- Line 171: How do you know that BC "was almost hydrophobic"? Did you measure it or do you assume it or are there previous measurements to back it up?

In this study, the hygroscopicity of BC was not measured. Based on the open literature, many studies have reported that BC is almost hydrophobic, for example, it has been reported that the hygroscopicity of BC particles displayed unimodal distribution, and their GF at 85% RH peaked at ∼ 1.0 (Li et al., 2018). However, it can also be hygroscopic when it mixed with other component or it acted as a core and coated by the secondary aerosols. When the ZSR method is applied to calculate the hygroscopicity of bulk aerosols with the assumption of external mixing state, BC was considered as a pure component and κ of 0 is commonly used, e.g. Pöhlker et al., 2023.

Li, K., Ye, X., Pang, H., Lu, X., Chen, H., Wang, X., Yang, X., Chen, J., and Chen, Y.: Temporal variations in the hygroscopicity and mixing state of black carbon aerosols in a polluted megacity area, Atmos. Chem. Phys., 18, 15201–15218, https://doi.org/10.5194/acp-18-15201-2018, 2018.

Pöhlker, M.L., Pöhlker, C., Quaas, J. et al. Global organic and inorganic aerosol hygroscopicity and its effect on radiative forcing. Nat Commun 14, 6139 (2023). https://doi.org/10.1038/s41467-023-41695-8

- Sect. 3.1 shows that there was something not working properly with the TSMPS system. It is not clear why the integrated values were significantly higher for the CF and CI cases, and why it suddenly agrees for the CR case. The total MCPC should always be similar or higher than the integrated values. The authors need to revise and present their detailed TSMPS set-up, check flows, PSL calibrations and zero-measurements (plus give details on the performed loss calculation, see comment above). Otherwise, the size distributions are questionable. My suspicion is that the diameter calibration is off in the Aitken-mode particle range, since it agrees better for the CR case. The PSL and high-voltage calibration are therefore key.

The high voltage and size calibration was conducted for TSMPS before the measurement started. In the LabView software, we use multi-point calibration to ensure that the input and output voltages exhibit a linear ratio, with a slope of 1250. For example, we input a voltage of 20 mv, then we measure the output voltage, it should be 25 v, otherwise, we adjust the slope in the software and do the calibration again. We use Latex of 200 nm to do the size calibration, to make sure DMA select the accurate monodispersed aerosols. If the measured particle size deviates from the PSL by 3%, the sheath flow rate will need to be adjusted. When the HV and size calibration have been conducted, we should believe the PNSD data are almost accurate. The aerosol and sheath flow, as well as the zero check are conducted regularly once a week.

The diffusion loss was considered in the date inversion program, with the input of tube length and flow rate (Wiedensohler et al., 2012). Number concentration of particles below 100 nm ($N_{<100nm}$) and above 100 nm ($N_{\geq 100nm}$) are obtained by integrating PNSD. It shows $N_{<100nm}$ with diffusion loss corrected is approximate 15% higher than the value without diffusion

loss correction. $N_{\geq 100nm}$ with diffusion loss corrected is approximate 2% higher, and almost no difference for particles above 200 nm (Fig. 3a). The difference between number concentration with diffusion loss corrected ($N_{diff,corr}$) and without ($N_{no,diff,corr}$) as indicated by the ratio of ($N_{diff,corr}$-$N_{no,diff,corr}$)/ $N_{diff,corr}$ shows significant size dependence. $N_{diff,corr}$ can by 70% higher than $N_{no,diff,corr}$ at 10 nm, and sharply decrease as the particle size increase (Fig. 3b). That means the diffusion loss can be ignored for particles above 100 nm. In this work, for the residual particles, which concentrate in the size above 100 nm, number concentration integrated from TSMPS ($N_t$) better agreed with that from MCPC ($N_{mcpc}$), with $N_t$ being 9% higher. For CF and CR particles, which was dominated by the Aitken mode particles, Nt was 30-40% higher than Nmcps, because the diffusion loss correction was conducted for TSMPS, but not for MCPC. The difference was much large for CF particles, as new particle formation event sometimes occurred, and the diffusion loss could be larger.

All the information about calibration is given in the supplementary materials.

[Figure]

Fig. s7. The comparison between mean PNSD of a day (288 scans) with diffusion loss correction and not (a), and size dependent difference of particle number concentration with corrected diffusion loss ($N_{diff,corr}$) and not $N_{no,diff,corr}$

Wiedensohler, A., Birmili, W., Nowak, A., et al., Mobility particle size spectrometers: harmonization of technical standards and data structure to

facilitate high quality long-term observations of atmospheric particle number size distributions, Atmos. Meas. Tech., 5, 657–685, https://doi.org/10.5194/amt-5-657-2012, 2012.

- Line 195: The wording "As expected" does not make sense. Please revise.

It has been corrected.

- Line 210 and Fig.3: The number concentration values for CR will most-likely go down if you include the sampling efficiency of the GCVI (see comment above).

The PNSD and mode fitting results for cloud residual particles in Fig.3 has been revised.

○ The measured values (solid lines) look very smooth. Did you apply any additional smoothing/averaging?

No, we do not use any smoothing method, just give the average PNSD under each condition, and the number of the samples was 2947, 1320 and 148 corresponding to cloud-free, cloud interstitial and residual particles, respectively. The mean and standard deviations of PNSD was given in the supplementary materials. Here only the mean value is given to make it easily reading because the mode fitting results are also given on the same plot.

○ For the CR cases, I would expect more variability also in the sub-100-nm range. Could you add standard deviations to your averages (or add a more detailed figure in the SI)?

The mean PNSD and standard deviation for cloud free, cloud interstitial and residual particles is given in the supplementary materials.

[Figure]

Fig. s8 The mean and median PNSD and its deviations for cloud free, cloud interstitial and residual particles.

- o The high values in the nucleation mode (~10nm) for the CF case are striking. Maybe this is also driven by some outliers, the authors could also try to use/show median values instead.

The high number concentration in nucleation mode for cloud free condition is contributed by the new particle events (NPF). There are six NPF cases observed for the entire measurement. The median PNSD was also given in the above figure.

- Line 218: Is the value by Karlsson et al. a mean or an estimate?

The number we referred from Kalsson et al., (2021) is the mean value. The original in Kalsson et al., (2021) is "The corresponding total particle

concentration during these cloud events is generally higher, ranging from 22 to 127 cm$^{-3}$ (25th and 75th percentiles) with a median of 55 cm$^{-3}$ (mean±SD: 101±143 cm$^{-3}$)."

- Line 239-240: What do you mean by "during these events, beyond the cloud process itself"? Are you suggesting that the cloud is leading to more interstitial aerosol during its presence? I think the confusing thing is that you call the size distribution measured behind the PM2.5 cyclone always CI (cloud interstitial) although there is no cloud present (see e.g. line 234). Suggest to revise this section and make the argumentation clearer.

I am sorry for the misunderstanding. In this sentence, we want to address that other source contributed to the increase of particles below 100 nm. Otherwise, there is no reason to explain the elevated number concentration during the cloud process. That also corresponded to the case study of May 8 in the following study, when polluted air mass arrived before cloud process, resulting in the higher number concentration of particles below 100 nm.

We defined cloud interstitial particles behind the PM$_{2.5}$ inlet during the cloud process, and cloud free particle behind the PM$_{2.5}$ inlet during cloud free condition. We have revised the sentences to make it clear.

- Figure 4: Please add to the caption the inlet behind which these PNSD were measured. I assume PM2.5? Also add if these are mean or median values.

The PNSD evolution pre-, onset and post cloud through the PM$_{2.5}$ inlet was given in Fig. 4. Before the half hour of cloud onset (T-0.5h), the mean PNSD is given for cloud free (CF) condition. The cloud interstitial PNSD when cloud process started and mean PNSD after cloud process within 1-3 hours until the PNSD slightly changes are also given. The figure has been revised, with the legend marked with CF (cloud free) and CI (cloud interstitial). The evolution of PNSDs during typical cloud episodes was analyzed (Fig. 4), including PNSDs corresponding to different stages:

before the cloud begins (30-minute average), cloud presence, cloud development and the mature stage (with 30-minute or 1-hour average) (a-d) and the corresponding size-resolved scavenging efficiency dependent on time (e-h). Although PNSDs showed significant variations before and during the cloud process and nucleation scavenging process was time dependent and, it was found once the cloud process was observed, the scavenging efficiency ($\eta$) tended to 1 for particles above the estimated activation diameter (marked by red box in e-h).

[Figure]

Fig. s9 The evolution of mean PNSDs half hour before the cloud onset and during the cloud processes, the scavenging efficiency about 0.5 was marked with the red box in e-h.

- Line 249: The authors state that Fig. 4 reveals that most significant changes in the PNSD are observed in the hour before the cloud event. However, this is not really evident from Figure 4. It is rather the opposite. The distributions change more after the event (red curves in Fig 4), while the time before the cloud started (T-0.5h) shows very similar PNSD as during the cloud.

We tried to illustrate the typical characteristics of PNSD evolution pre-cloud, onset and post-cloud processes, especially highlighted how the cloud process modifies the PNSD. The onset of cloud is detected by GCVI, with setting threshold of visibility below 1 km and RH above 95%. Once

the GCVI started, the inlet will switch between GCVI (cloud residual) and PM$_{2.5}$ (cloud interstitial) automatically. As our GCVI and PM$_{2.5}$ inlet system can capture the PNSD evolution with high time resolution (5 min for each TSMPS scan) for the entire cloud process, especially the onset stage of cloud, it facilitates the comparison of PNSD between cloud free and cloud interstitial particles, thus, we can study how the particles are activated into cloud droplets. In this work, we applied the comparison of mean PNSD between half an hour before the onset of cloud (cloud free particles) and half an hour after onset of cloud (cloud interstitial) to determine the particles which were activated as cloud droplets. As part of the aerosols can be activated as cloud droplet and scavenged from the aerosol system, the reduction of particle number concentration can be found in the changes of PNSD. We focused on the transition period between CF and CI particles, in order to reveal how the aerosol are activated into cloud droplets. It was found the critical diameter normally larger than 100 nm in this study. For the particles below 100 nm, they can not be activated probably because the super saturation is not enough. For the further changes of PNSD in 1-3 hours after the cloud formation, the particle larger than the activated diameter reduced further due to the nucleation scavenging, while the particles below 100 nm could be even higher, probably related with the changes of air mass.

- Line 251: How can you determine the critical activation diameter from "the evolution PNSDs throughout the cloud process"? This needs more explanation.

In this work, we applied the comparison of mean PNSD between half an hour before the onset of cloud (cloud free particles) and half an hour after onset of cloud (cloud interstitial) to determine the particles which were activated as cloud droplets. As part of the aerosols can be activated as cloud droplet and scavenged from the aerosol system, the reduction of particle number concentration can be found in the changes of PNSD. We focused on the transition period between CF and CI particles, in order to reveal how the aerosols are activated into cloud droplets. The particle size

corresponding to the scavenge efficiency of 50% was defined as the critical activation diameter.

- Sect. 3.3. and Fig.5: The scavenging efficiency values presented here are still questionable since the authors have not determined the (time-dependent!) cloud droplet sampling efficiency of their GCVI (see comment above). This needs to be done first and then the calculations need to be repeated. Although, Figure 5 indicates that the sampling efficiency reaches ~1 on average at larger diameters, I wonder, why? Did you scale the CR PNSD data? If so, this needs to be described! If you did not scale, then it would mean that the larger droplets (which are usually poorly sampled by the GCVI) are not important. However, this would need to be determined case by case. The authors should make much more use of their measured cloud droplet spectra. As will be shown later (Fig 8b and 9b), there is indeed quite a dominance of small droplets around a few micrometers, but the variability is large. The final PNSD from the GCVI inlet (CR) should match the ambient droplet concentration (see Karlsson et al. 2021).

In fig. 5, the comparison between cloud free particles (before the cloud process) and cloud interstitial particles (the initial stage of cloud presence) was conducted to derive the scavenging efficiency and activation behaviors. The scavenging efficiency ($\eta$) was defined as $\eta = \frac{C_{pre,cf} - C_{ci}}{C_{pre,cf}}$ , equation (1) in the manuscript and the calculation didn't consider the cloud residual particles by GCVI. The large value of $\eta$ , close 1, indicating the number concentration of cloud interstitial particles is tending to zero, and all the particles are activated as cloud droplets. So in Fig. S9, the larger particles with $\eta$ , close to 1, indicated these particles are activated as cloud droplets. We are agree with the reviewer's comment that the scavenging efficiency is time dependent. So we revised figure 4 that average 30 min PNSD of cloud interstitial particles is divided by the cloud free particles before the

presence of cloud passage, to evaluate the time evolution of PNSD changes of interstitial particles.

- Line 256-258: "…are likely removed by coagulation processes within the cloud". How does this work? Isn't it just measurement uncertainty and potential changes in ambient aerosol concentration (and their corresponding PNSD)?

We agreed with the reviewer's comment that measurement uncertainties exit in the PNSD measurement, especially for the ultrafine particles below 100 nm, due to diffusion loss in the inlet and tube pipes. And the changes of ambient aerosols also reshape the PNSD, such as the changes of air mass as we have mentioned. Here, we just want to express the influence of cloud process on the PNSD, for example, the ultrafine particles can be solved or coagulated by the cloud droplet. We have revised this sentence, and supplemented other possibility as reviewer suggested.

- Line 260: The entrainment argument is very speculative. Do you have any evidence for this?

This sentence has been removed.

- Line 261-263: Shouldn't it be "before and during cloud" since you used the CR (GCVI PNSD) to determine the activation diameters?

This sentence has been revised to "This observation implies that the evolution in PNSDs before (CF particles) and during the cloud process (CI particles, particularly in the initial stage of cloud formation) can serve as an indicator of the critical diameter for particles activation into cloud droplets under ambient conditions."

- Line 265-267: At the end of the sentence, you should better say: "changes in air mass during the cloud event". This is one of the most crucial assumptions you are making in your approach.

Thanks for the comments, the sentence has been revised accordingly.

- Line 269-270: It depends also on the particle number concentration and the mixing state of the respective ambient aerosol.

The sentence has been revised to "…depends on the particle number size distribution, hygroscopicity, mixing state, and the supersaturation of the cloud.

- Line 285-287: The authors hypothesize that the ultrafine particles take up the water since they "can be also hygroscopic" and thus inhibit larger particles to grow. This is based on the determined activation diameter (see Fig. 5), which shows quite some uncertainties.

We agree with the reviewer's comments. The activation of aerosols into cloud droplets is determined by aerosol properties and the super saturation. If we only address the particle hygroscopicity, it is one-side view. So we remove this sentence.

- Line 287-289: The LWC values for the different cloud events are not shown. I would suggest that you include a summarizing table with all the relevant parameters (LWC, activation diameters, kappa values, chemical composition, etc.) to the revised manuscript (also including the variability of the different parameters).

The whole time series of fog micro-physical properties and the key parameters including the duration time, number concentration of cloud droplet ($N_d$), liquid water content (g m$^{-3}$), effective diameter ($D_{pe}$), activated diameter ($D_c$), hygroscopicity parameter ($\kappa$) and mass concertation for cloud free, cloud interstitial and residual particles are given, also in the supplemented materials. In this study, we only focused on the cases of April 19, 28, May 5 and 8, with available PNSD data and without precipitation. Unfortunately, the fog data on May 5 is not available.

[Figure]

Fig.s10 Time series of (a) fog droplets size distribution, (b) liquid water content (*LWC*), and effective diameter ($D_{pe}$), and (c) fog droplet number concentration ($N_{fog}$) and precipitation

Table 1. Key parameters including the duration time, number concentration of cloud droplet ($N_d$), liquid water content (g m$^{-3}$), effective diameter ($D_{pe}$), activated diameter ($D_c$), hygroscopicity parameter ($\kappa$) and mass concertation for cloud free, cloud interstitial and residual particles

| Cloud episode | $N_d$ (cm$^{-3}$) | LWC (g m$^{-3}$) | $D_{pe}$ (μm) | $D_c$ (nm) | $\kappa$ | | | Mass concentration ( μg m$^{-3}$) | | |
|---|---|---|---|---|---|---|---|---|---|---|
| | | | | | CF | CI | CR | CF* | CI | CR |
| April 19 16:30-23:00 | 629±304 | 0.24±0.08 | 11.7±2.6 | 325 | 0.27±0.01 | 0.23±0.06 | 0.34±0.04 | 10.6±1.2 | 7.3±4.3 | 6.5±2.5 |
| April 28 12:10-23:00 | 388±107 | 0.25±0.06 | 12.9±1.5 | 159 | 0.30±0.01 | 0.25±0.03 | 0.29±0.02 | 9.8±0.6 | 1.4±1.2 | 1.6±0.5 |
| May 5 18:50-May 6 04:00 | - | - | - | 133 | 0.28±0.01 | 0.27±0.02 | 0.30±0.03 | 5.3±0.1 | 2.5±1.6 | 2.3±1.2 |
| May 8 19:10-22:00 | 771±263 | 0.04±0.02 | 5.5±0.5 | 199 | 0.29±0.01 | 0.26±0.03 | 0.32±0.01 | 17.6±0.5 | 10.9±7.9 | 15.5±1.8 |

CF, CI and CR indicates the cloud free, cloud interstitial and cloud residual particles

*The mean and standard deviation was calculated for the 30 min before cloud presence as cloud free particles

-indicates the data are not available

- Paragraph starting in Line 293: Here the authors determine the number concentration of CCN probably using the PNSD measured behind the GCVI and compare it to the CCNC spectra (Fig. 6). First of all, it is not really clear which PNSD were used. If the GCVI data (CR) was used, then you need to include the GCVI sampling efficiency and cut-off diameter, since you only sample a sub-set of the droplet distribution! In addition, why don't you just use the (corrected) fog monitor data? This will probably lead to different supersaturation values. In any case, you need to thoroughly also address the uncertainties in your calculations!

The estimation of CCN number concentration was calculated based on the critical diameter ($D_c$, determined by the comparison between PNSD before and during the cloud process) and PNSD before cloud. The particles number concentration from Dc to the upper size limit of PNSD was integrated and defined as CCN concentration. We used only CF and CI PNSD, without CR particles. The fog data are only available on April 28 and May 8, which is too limited to yield statistically significant results.

- Line 310: the PM1 mass concentration will change if you include the sampling efficiency and cut-off diameter of your CR data.

The mass concentration of $PM_1$ and the figure of chemical component fraction have also been corrected, with considering the GCVI sampling efficiency of 0.68.

- Line 311: "CI particles had the highest organics mass fraction". This is not evident from Fig. 7. CI and CR show almost the same organic mass fractions and fluctuations are within error margins. What is evident, is that the black carbon fraction is clearly higher in the CI particles (leading to a lower kappa) compared to the CR and CF cases. This you could discuss as well. A table with mean and standard deviation values for all the fog events would be more convincing here (see comment above).

We revised the discussion here. The mass fraction of organics doesn't show a clear variation for CF, CI and CR particles, approximately 40%. The higher mass fraction of eBC in CI particles, corresponding to a lower κ. That indicated the hydrophobic particles are difficult to be activated into cloud droplets, which are retained in the interstitial particles. A table with mean and standard deviation values for all the fog events has been supplemented.

- Line 329-334: If I understood correctly, you calculated the scavenging efficiencies using the observations before and during the cloud. How did the actual concentrations change/decrease after the cloud? Were the constituents really removed or just activated and then released again?

Yes, in the present discussion, we only focused on the before and the presence of cloud processes. Based on our experiment, we found after the cloud processes, the particle number concentration and mass concentration also changed significantly, depending on the air mass transport. For example, on May 8, the air masses changed to more polluted regions (from Central China) and resulted in elevated particle concentration since before the presence of cloud process. So although the part of particles are scavenged during the cloud processes, the particle concentration increased immediately after the cloud passage. We also supplemented the discussion in the manuscript.

- Sect. 3.4: This is a nice case study. Ideally you can present it first, before you come to your general results.

The authors have re-organized the paper structure as the reviewer suggested.

- Line 350: In addition to the trajectories, indicating "significant air pollutants" being transported to the site, you could actually show

it but presented to corresponding AMS and MAAP data in Fig. 8. Right now, it is just indicative.

The time series of chemical composition mass concentration and the mass fraction, including organics, nitrate, sulfate, ammonium and equivalent black carbon was supplemented. It showed the mass concentration started to increase since 15:00 LT, as influenced by the air mass change. The particle mass concentration sharply decreased as the cloud process (19:00-22:30), but the mass fraction can reflect the relative variation of each component. As cloud presence, the mass fraction of sulfate decreased, whereas nitrate mass fraction increased. The cloud episode before 6:00 LT was not discussed because it was influenced by the precipitation on May 7 and the both CI and CR particle number concentration remained low.

[Figure]

Fig. s1 Particle number size distribution (contour plot), as well as relative humidity (blue line), visibility (black line) and inlet system state (red) on May 8, (b) number size distribution of cloud droplet, liquid water content (purple line), geometric mean diameter, $D_{pe}$ (black line) and (c) mass concentration of chemical composition (organics, nitrate, sulfate, ammonium and equivalent black carbon).

- Line 357: How do you know that cloud formation occurred? It could also be that the air mass was changing and cloudy air was transported to the site. At a stationary site, clouds are probably always a moving process and an interplay between meteorology, transport and aerosol properties. I would write here and within the entire section "cloud presence" or "residence within cloud air"

It has been revised to "showed that the presence of cloud passage was observed around 19:15 LT"

- The droplet spectra in Fig. 9b is interesting. One can not really see a distinct droplet mode (especially at the beginning of the low visibility). How do you know that these are activated cloud droplets and not hydrated aerosols (haze)? The later distributions at 19:35 and 19:40 could indicate a small droplet mode at around 25 micrometers. You could show a composite of dry and wet PNSD (from the TSMPS) and the droplet distributions (from the FM).

- In addition, as mentioned above, have you corrected the FM for losses (important especially for the larger droplets)? How was the wind speed at the event and how was the FM orientated (this info should be added to the method part). Especially the updraft velocity would be interesting to compare to your case study shown in Fig. 9.

Hydrated aerosols of the accumulation mode co-existed with droplets, as interstitial non-activated aerosols. Their size continued to increase, and some aerosols achieved diameters larger than 2.5 µm. In the previous study, it has been reported that the mean transition diameter between the aerosol accumulation mode and the small droplet mode was $4.0 \pm 1.1$ µm (Elias et al., 2009; 2015). The contribution of interstitial particles to the light extinction can not be ignored (Liu et al., 2024). For the hydrated aerosols larger than 2.5 µm, they will be removed by the impactor and can not enter the inlet system, which resulted in the underestimation of cloud interstitial

particles. However, this part of hydrated aerosol can be detected by FM and mistaken as cloud droplets. Unfortunately, it is difficult to differentiate these aerosols quantitively. But in this case, the small droplet mode around 2.5 µm by FM, also corresponded to the highest concentration of LWC, indicating the contribution by cloud droplets is overwhelming. We also supplemented the uncertainty in the manuscript.

The FM sampling efficiency was considered, with a corrected factor of 0.95. During this measurement, the wind speed is 5.8±2.6 m/s. FM is put on the roof of cabinet, with the inlet direction to the downtown at the foothill in the northeast.

Elias, T., Haeffelin, M., Drobinski, P., Gomes, L., Rangognio, J., Bergot, T., Chazette, P., Raut, J.-C., and Colomb, M.: Particulate contribution to extinction of visible radiation: Pollution, haze, and fog, Atmospheric Research, 92, 443-454, 505, 2009.

Elias, T., Dupont, J.-C., Hammer, E., Hoyle, C. R., Haeffelin, M., Burnet, F., and Jolivet, D.: Enhanced extinction of visible radiation due to hydrated aerosols in mist and fog, Atmos. Chem. Phys., 15, 6605 – 6623, https://doi.org/10.5194/acp-15-6605-2015, 2015.

- Sect 3.5: This section needs some thorough revisions. Right now, I am not learning much new here and in parts I am more confused after reading. Many of the statements made here can only be done if the meteorological conditions are similar, which is not easy due to the limited extent of the dataset.

  - First of all, are the scatterplots in Figure 10 just for the cloud event on the 18th of April (as mentioned in the first paragraph) or for the entire period?

  - Second, and most importantly, some of the statements are very hand-waving and I would recommend focusing on the clear relationships. There is no relationship between the geometric diameter of the CI and CR PNSD and the cloud parameters. So maybe remove it (or move

it to the SI)? Also, keep in mind that LWC and Dpe are correlated (larger or growing droplets = more LWC). I would make this point clear.

- o Line 393-395: The inverse relationship between LWC and kappa is indeed interesting. However, changes in meteorology are also a possible explanation. Can you rule out effects of temperature, updraft or air mass change?

- o Third row in Fig 10: I doubt that the N130 is also relevant for the CF, since you have particles smaller than 130 nm that have activated (see Fig. 3), so better replace it with the total number concentration and repeat the analysis (see also comments above).

- o Line 407: There is an inverse relationship between f_N130 and LWC and D_pe, but not between f_N130 and Nd (Fig 10k).

We modified Fig. 10 and only May 8 case was studied. The scatter plot of particle hygroscopic parameter ($\kappa$), geometric mean diameter ($D_{pg}$), number concentration from MCPC ($N_{mcpc}$) for cloud interstitial, CI and cloud residual (CR) particles as well the scatter plots of $\kappa$ and cloud droplet parameters (LWC, $N_d$, and $D_{pe}$) for CI and CR are analyzed to support the conclusion about how aerosol properties influenced the cloud droplets,

[Figure]

Fig. 11 The scatter plots of particle hygroscopic parameter ($\kappa$), geometric mean diameter ($D_{pg}$), number concentration from MCPC ($N_{mcpc}$) for cloud interstitial, CI (a) and cloud residual (CR) particles (b), as well the scatter plots of $\kappa$ and cloud droplet parameters (LWC, $N_d$, and $D_{pe}$) for CI (c) and CR (d) on May 8 case, respectively.

- Within the conclusions, you need to make clear that these are only a few case studies (less than 24 hours of in cloud data over 4 days) and that the generalization of the results are difficult to make since they all depend on meteorological conditions which are not properly addressed here due to the given limitations.

We have revised the conclusion part and discuss the limitations of this study.

- Line 443: The uncertainty of 10% is a wild guess, since you have not even performed the loss calculations for the FM and due the fact that the transition between hydrated aerosol and droplets seems to be a continuous regime (see Fig 9b and comments above).

We removed the quantitative estimation of the discrepancy between cloud droplets and particles probably activated as cloud droplets. We also discussed the uncertainty that could be introduced by the hydrated particles.

- Line 451: How much higher was it? Could you add some numbers here? The higher CI particle number concentrations could also be due to changes in air mass.

The number concentration of ultrafine particles below 100 nm ($N_{dp<100nm}$) was integrated from PNSD for cloud free and interstitial particles, it was found for April 19 and May 8 cloud processes, $N_{dp<100nm}$ was approximately 50%, or even higher for CI particles, as compared with that for CF particle before the presence of cloud. That indicated other sources contributing to the ultrafine particles, such as the long-range transport. It has been also revised in the manuscript.

**Minor comments:**

- Line 62: "physic-chemical" -> "physico-chemical"

It has been corrected.

- Fig. 3 caption: Better say "shown for the entire measurement period." And state that the fits are log-normal fits.

It has been corrected.

- Within the manuscript, the authors often use the term "cloud processes" but just mean that they are measured during the presence of clouds.

We have revised to "the presence of cloud passage was observed"

- Line 346: incoming -> changing (air has hopefully been always around)

It has been corrected.

- Line 348: are -> were

It has been corrected.

- Line 391: It should be Fig 10.

It has been corrected.

- Line 453: Check typos (mechanisms, remains)

It has been corrected.

**References:**

Adachi K, Tobo Y, Koike M, Freitas G, Zieger P, Krejci R. Composition and mixing state of Arctic aerosol and cloud residual particles from long-term single-particle observations at Zeppelin Observatory, Svalbard. Atmospheric Chemistry and Physics. 2022 Nov 10;22(21):14421-39.

Duplessis P, Karlsson L, Baccarini A, Wheeler M, Leaitch WR, Svenningsson B, Leck C, Schmale J, Zieger P, Chang RW. Highly Hygroscopic Aerosols Facilitate Summer and Early-Autumn Cloud Formation at Extremely Low Concentrations Over the Central Arctic Ocean. Journal of Geophysical Research: Atmospheres. 2024 Jan 28;129(2):e2023JD039159

Karlsson L, Krejci R, Koike M, Ebell K, Zieger P. A long-term study of cloud residuals from low-level Arctic clouds. Atmospheric Chemistry and Physics. 2021 Jun 14;21(11):8933-59.

Pereira Freitas G, Kopec B, Adachi K, Krejci R, Heslin-Rees D, Yttri KE, Hubbard A, Welker JM, Zieger P. Contribution of fluorescent primary biological aerosol particles to low-level Arctic cloud residuals. Atmospheric Chemistry and Physics. 2024 May 13;24(9):5479-94.

Spiegel JK, Zieger P, Bukowiecki N, Hammer E, Weingartner E, Eugster W. Evaluating the capabilities and uncertainties of droplet measurements for the fog droplet spectrometer (FM-100). Atmospheric Measurement Techniques. 2012 Sep 20;5(9):2237-60.

Zieger P, Heslin-Rees D, Karlsson L, Koike M, Modini R, Krejci R. Black carbon scavenging by low-level Arctic clouds. Nature communications. 2023 Sep 7;14(1):5488.

We appreciate the reviewer for providing valuable references, which we have included in the reference list.